# Mechanism of Huanglian Wendan Decoction in ameliorating non-alcoholic fatty liver disease via modulating gut microbiota-mediated metabolic reprogramming and activating the LKB1/AMPK pathway

Jianping Zhu[1,2], Yuzhen Chen[2], Yidi Han[1], Ji Li[1]*

**1** School of Basic Medicine, Heilongjiang University of Chinese Medicine, Harbin, China, **2** Pharmacy of College, Hunan University of Chinese Medicine, Changsha, China

* lijihljfj@126.com

## Abstract

### Background

Huanglian Wendan Decoction (HLWDD), a classical traditional Chinese medicine (TCM) formula, has shown therapeutic promise in treating metabolic disorders. However, its underlying mechanisms against non-alcoholic fatty liver disease (NAFLD) remain unclear.

### Objective

This study aimed to elucidate the pharmacological mechanisms by which HLWDD ameliorates NAFLD, focusing on its impact on lipid metabolism, gut microbiota, and amino acid regulation.

### Methods

A NAFLD rat model was established by administering a high-sugar, high-fat, high-salt diet for 20 weeks. The core components of HLWDD were identified and quantified using UPLC-Q-TOF-MS/MS and HPLC, and further validated via network pharmacology and molecular docking. Therapeutic efficacy was assessed through analysis of body weight, serum lipid profiles, inflammatory cytokines, hepatic histology, and protein expression. Gut microbiota composition and liver-intestine metabolite profiles were evaluated using metagenomic sequencing and LC-MS/MS.

### Results

Seven key constituents, including quercetin and berberine, were quantified (15.11–164.37 µg/mL) and shown to interact with lipid metabolism targets such as liver

**Data availability statement:** All relevant data are within the manuscript and its Supporting Information files.

**Funding:** This work was supported by Hunan Provincial Department of Education Research Project (23B0375); Hunan Provincial Administration of Traditional Chinese Medicine Research Project (C2023021); Hunan University of Traditional Chinese Medicine University-level Research Project (2024XJZC017).

**Competing interests:** The authors declare no conflict of interest.

**Abbreviations:** AMPK, AMP-activated Protein Kinase; CPT1A, Carnitine Palmitoyltransferase 1A; HDL, High Density Lipoprotein; H&E, Hematoxylin and Eosin; HPLC, High Performance Liquid Chromatography; HLWDD, Huanglian Wendan Decoction; IL, Interleukin; LDL, Low Density Lipoprotein; LKB1, Liver Kinase B1; NAFLD, Non-Alcoholic Fatty Liver Disease; NF-κB, Nuclear Factor kappa-B; PPARα, Peroxisome Proliferator-Activated Receptor Alpha; PPARγ, Peroxisome Proliferator-Activated Receptor Gamma; SCFAs, Short-Chain Fatty Acids; TC, Total Cholesterol; TCM, Traditional Chinese Medicine; TG, Triglyceride; TNF-α, Tumor Necrosis Factor-α; UPLC-Q-TOF-MS/MS, Ultra-Performance Liquid Chromatography Quadrupole-Time-of-Flight Tandem Mass Spectrometry.

kinase B1 (LKB1), AMP-activated protein kinase (AMPK), peroxisome proliferator-activated receptor alpha (PPARα), and carnitine palmitoyltransferase 1A (CPT1A). HLWDD treatment significantly reduced body weight, hepatic lipid accumulation, and serum levels of triglycerides, total cholesterol, and low-density lipoprotein cholesterol, while increasing high-density lipoprotein cholesterol. Proinflammatory cytokines (IL-6, IL-1β, TNF-α) were notably suppressed. Mechanistically, HLWDD activated the LKB1/AMPK signaling pathway and modulated aspartic acid metabolism in association with increased abundance of *Akkermansia* in the gut. Metabolomic analysis identified 13 differential metabolites, with aspartic acid showing strong correlations with *Akkermansia* and LKB1/AMPK activity.

## Conclusion

HLWDD exerts its anti-NAFLD effects by enhancing *Akkermansia*-mediated aspartate metabolism, thereby activating the LKB1/AMPK axis and promoting lipid oxidation via CPT1A and PPARα. This study provides new mechanistic insight into the gut–liver axis in NAFLD and highlights HLWDD as a multi-targeted therapeutic approach for restoring metabolic balance.

## Introduction

NAFLD, the most prevalent chronic liver disorder globally, affects over 25% of the population and is strongly linked to obesity, type 2 diabetes, and metabolic syndrome [1–3]. Its hallmark pathological feature—excessive lipid deposition in hepatocytes—arises from dysregulated energy metabolism, gut microbiota dysbiosis, and gut-liver axis impairment [4–6]. The LKB1/AMPK signaling axis plays a central role in hepatic lipid homeostasis. Phosphorylation of AMPK (Thr172) by LKB1 inhibits fatty acid synthesis while promoting CPT1A/PPARα-driven oxidation [7–9]. Conversely, LKB1 deficiency or AMPK inactivation exacerbates gluconeogenesis and oxidative stress, accelerating NAFLD progression [10]. Although AMPK activators show therapeutic potential, concerns regarding tissue specificity and long-term safety persist [10].

The gut microbiota critically interacts with host metabolism via the gut-liver axis. NAFLD patients exhibit characteristic dysbiosis, often marked by reduced beneficial bacteria [11–13]. Dysregulated gut-derived metabolites (e.g., LPS, TMAO) exacerbate hepatic lipid accumulation and insulin resistance [11–13]. While probiotics partially alleviate NAFLD, inter-individual variability and microbiota instability hinder clinical translation [14,15].

TCM offers promising multi-target therapeutic strategies for complex diseases like NAFLD. HLWDD, a classical heat-clearing formula, has demonstrated efficacy in improving dyslipidemia, insulin resistance, and hepatic inflammation [16,17]. Evidence suggests HLWDD reduces hepatic steatosis by suppressing inflammatory pathways (e.g., PPARγ/NF-κB) and modulates gut microbiota, increasing beneficial metabolites like short-chain fatty acids (SCFAs) [16,18]. However, despite evidence

for the metabolic and microbiota-modulating effects of HLWDD constituents, the precise mechanisms through which HLWDD influences the gut-liver axis and AMPK signaling in NAFLD remain unclear.

Therefore, this study aimed to address the specific question: Does HLWDD ameliorate NAFLD through modulation of the gut-liver axis (specifically *Akkermansia*-dependent aspartate metabolism) and activation of the LKB1/AMPK pathway? We hypothesize that HLWDD alleviates NAFLD by enriching beneficial gut flora (e.g., *Akkermansia*), regulating gut-liver axis metabolites to activate LKB1/AMPK signaling, and restoring lipid oxidation (via CPT1A/PPARα) and amino acid homeostasis. This study will elucidate the synergistic effects of HLWDD on microbial metabolites and hepatic energy metabolism, providing novel mechanistic insights and a potential therapeutic avenue for NAFLD.

## Materials and methods

### Reagents and chemicals

TE buffer (Cat. No. R541019-0100), ELISA kits for IL-6 and IL-1β (Batch No. GR20230610), TNF-α (M130055-48T), triglyceride (TG), HDL-C, and LDL-C assay kits were purchased from commercial suppliers. RNA extraction kit (Cat. No. 5003050), SYBR qPCR SuperMix (E099-01A), and cDNA synthesis mix (E047-01B) were used for gene expression analysis. Primers for AMPK, LKB1, CPT1A, PPARα, and internal control (ACTB) were obtained from Sangon Biotech (Shanghai) Co., Ltd. Primary antibodies (p-LKB1, p-AMPK, LKB1, AMPK, PPARα, CPT1A, GAPDH) were sourced from Abcam. Chromatographic-grade solvents (methanol, acetonitrile, formic acid, water, isopropanol) were obtained from Fisher Scientific. Metformin (catalog: 20200A) was obtained from Patheon France. Reference standards of HLWDD compounds (berberine, palmatine, quercetin, etc.) were purchased from Sichuan Cuiyirun Biotechnology.

### Preparation of HLWDD

Herbal ingredients with specified botanical sources and plant parts were weighed: *Coptis chinensis* Franch. (rhizome; Sichuan, China; Batch: CC202305) 10g, *Pinellia ternata* (Thunb.) Makino (tuber; Shanxi, China; Batch: PT202304) 10g, *Phyllostachys edulis* (Carrière) J.Houz. (caulis; Zhejiang, China; Batch: PE202306) 10g, *Citrus aurantium* L. (immature fruit; Jiangxi, China; Batch: CA202302) 15g, *Citrus reticulata* Blanco (pericarp; Guangdong, China; Batch: CR202303) 10g, *Glycyrrhiza uralensis* Fisch. Ex DC. (root; Inner Mongolia, China; Batch: GU202307) 5g, *Zingiber officinale* Roscoe (fresh rhizome; Shandong, China; Batch: ZO202305) 5g, *Wolfiporia cocos* (F.A.Wolf) Ryvarden & Gilb. (sclerotium; Yunnan, China; Batch: WC202308) 15g. All specimens were morphologically authenticated by Prof. Tasi Liu (School of Pharmacy, Hunan University of Chinese Medicine) and taxonomically verified using Medicinal Plant Names Service (MPNS; http://mpns.kew.org). Voucher specimens were deposited in the University Herbarium (Accession No. HLWD2023−001–008).

Decoction was performed in two steps: First decoction: add 8-fold water, boil for 1 h, filter. Second decoction: add 6-fold water, boil for 30 min, filter. Combine filtrates, concentrate to 100 mL. Mix 10 mL of decoction with ethanol to form 70% solution, refrigerate for 12 h, then filter. Dilute 2 mL filtrate with methanol, mix with internal standard, filter through 0.45 μm membrane, and centrifuge at 15,000 rpm for 5 min.

### Quantification of core components

Reference compounds were weighed, dissolved in methanol, and diluted into calibration solutions. HPLC analysis was performed using Elite C18 column (4.6×250 mm, 5 μm), with a mobile phase of acetonitrile (A) and 0.1% ammonium acetate (B). The gradient was 10% A to 50% A over 40 min. Detection was at 280 nm, flow rate 0.8 mL/min, column temperature 30°C.

### Network pharmacology and molecular docking

Candidate compounds from HLWDD were identified using UPLC-Q-TOF-MS/MS. Targets were retrieved from TCMSP and SwissTargetPrediction, standardized via UniProt, and NAFLD-related targets identified from GeneCards. Venny 2.1.0 was

used to find overlapping targets. PPI networks were constructed with STRING and analyzed with Cytoscape (v3.9.0). GO and KEGG enrichment analyses were conducted using DAVID. Molecular docking was performed using AutoDock 4.2.6 and visualized with PyMol.

## Animal experimentation

Animal experiments were approved by the Animal Ethics Committee of the Hunan University of Chinese Medicine (Approval No. LLBH-LL2022111704) and conducted in accordance with the Guide for the Care and Use of Laboratory Animals. A total of 64 male SPF-grade Sprague-Dawley rats ($200 \pm 20$ g) were housed (3/cage, $20°C \pm 2°C$, 12 h light/dark cycle) with ad libitum access to water and standard chow. After 1 week of adaptation, 8 rats formed the control group (normal diet), and 56 received a high-fat, high-sugar, high-salt diet for 20 weeks to induce NAFLD. Successfully modeled rats were randomized into five groups (n = 8/group): model (vehicle), metformin (90 mg/kg) [19], and HLWDD low (3.6 g/kg), medium (7.2 g/kg), and high (14.4 g/kg) dose groups. The selection of HLWDD doses was based on clinical equivalent dose conversion. According to FDA guidelines for dose translation from animal to human studies [20], the human-to-rat equivalent dose ratio is 6.25:1 based on body surface area normalization. Given the clinical dosage of HLWDD in adults is 2.271 g·kg$^{-1}$·d$^{-1}$ [21], the calculated rat equivalent dose is 14.196 g·kg$^{-1}$·d$^{-1}$ (rounded to 14.4 g/kg). Medium and low doses (7.2 and 3.6 g/kg) were set at 50% and 25% of the high dose, respectively, to evaluate dose-response relationships. All efforts were made to minimize suffering; body weight, food intake, and clinical signs of distress were monitored daily, and veterinary staff were available. No animals required early euthanasia due to distress. After 4 weeks of oral treatment, rats were fasted for 12 h, deeply anesthetized via intraperitoneal injection of sodium pentobarbital (50 mg/kg) (unresponsiveness confirmed by pedal reflex), and euthanized by exsanguination via cardiac puncture. Death was confirmed by cessation of heartbeat and respiration. Blood and liver tissues were collected for analysis. All animals completed the study without mortality or attrition.

## Histological and biochemical analysis

Liver tissues were fixed in 4% paraformaldehyde and embedded in paraffin. Sections (5 µm) were stained with H&E and examined by light microscopy. Serum lipid profiles (TC, TG, HDL-C, LDL-C) and liver cytokines (TNF-α, IL-1β, IL-6) were measured by commercial kits.

## Gene and protein expression

RNA was extracted via TRIzol and reverse transcribed to cDNA. qPCR was performed using SYBR Green with β-actin as reference. Expression of AMPK, LKB1, CPT1A, PPARα, etc., was calculated using the 2-ΔΔCt method. The sequences of primers are shown in Table 1. Protein expression was analyzed by Western blot. Liver proteins were lysed, separated via SDS-PAGE, transferred to PVDF membranes, and probed with primary/secondary antibodies. Detection was via ECL and quantified with Image *J*.

**Table 1. Real-time PCR primer sequence.**

| Gene | Uterus sequence (5'→3') | Cells sequence (5'→3') |
|---|---|---|
| AMPK | TGATGAGGTGGTGGAGCAGAG | AGTGAGAGAGCCAGACAGTGAATG |
| LKB1 | ACACCTTCATCCACCGCATCG | GTCCAGCACCTCCTTCACCTTG |
| CPT1A | CAGGAGAGTGCCAGGAGGTCATAG | TGCCGAAAGAGTCAAATGGGAAGG |
| PPARα | TCTTCACGATGCTGTCCTCCTTG | TGTCGCAGAATGGCTTCCTCAG |

## Microbiome and metabolomic analysis

Fecal DNA was extracted using PF Mag-Bind Stool DNA Kit. DNA integrity was assessed by agarose gel and quantified by NanoDrop. Metagenomic reads were assembled by MEGAHIT, ORFs predicted with Prodigal, and gene abundance analyzed using SOAPaligner.

Metabolites were analyzed using a Thermo UHPLC-Q Exactive HF-X system with ACQUITY HSS T3 column. Positive and negative ion mode separation was performed with defined gradients. Flow rate: 0.40 mL/min; column temperature: 40°C.

## Statistical analysis

Data were analyzed using GraphPad Prism 9 and SPSS 24.0. Results are expressed as mean ± SD. One-way ANOVA followed by LSD-t post hoc test was applied. $P < 0.05$ was considered statistically significant. KEGG-based pathway enrichment was performed via Python's scipy.stats and Fisher's exact test.

## Results

### Ingredient identification analysis

UPLC-Q-TOF-MS analysis of HLWDD yielded a total ion current chromatogram (Figs 1 and 2). A total of 58 chemical components were identified based on MS/MS fragmentation patterns and database matching using ChemSpider and MassBank; detailed information, including retention times and MS/MS spectra, is provided in S1 Table. These included two amino acids and derivatives (1, 3), two nucleosides and derivatives (2, 58), seven phenolic acids (5, 6, 8, 9, 11, 18, 23), twenty-five flavonoids and glycosides (12–17, 22, 24, 26, 27, 30, 33, 34, 36–38, 40–44, 49, 53, 54), six alkaloids (7, 19, 25, 29, 32, 52), one terpenoid (35), eight coumarins and furanocoumarins (31, 39, 45, 46, 50, 51, 55, 57), two fatty acids and carboxylic acids (4, 28), and five other compounds (20, 21, 47, 48, 56).

### Network pharmacology analysis

Network pharmacology identified 229 shared targets between HLWDD active ingredients and NAFLD (Fig 3). These are listed in S2 Table for transparency. Among them, 30 potential core targets were used to construct a PPI network in

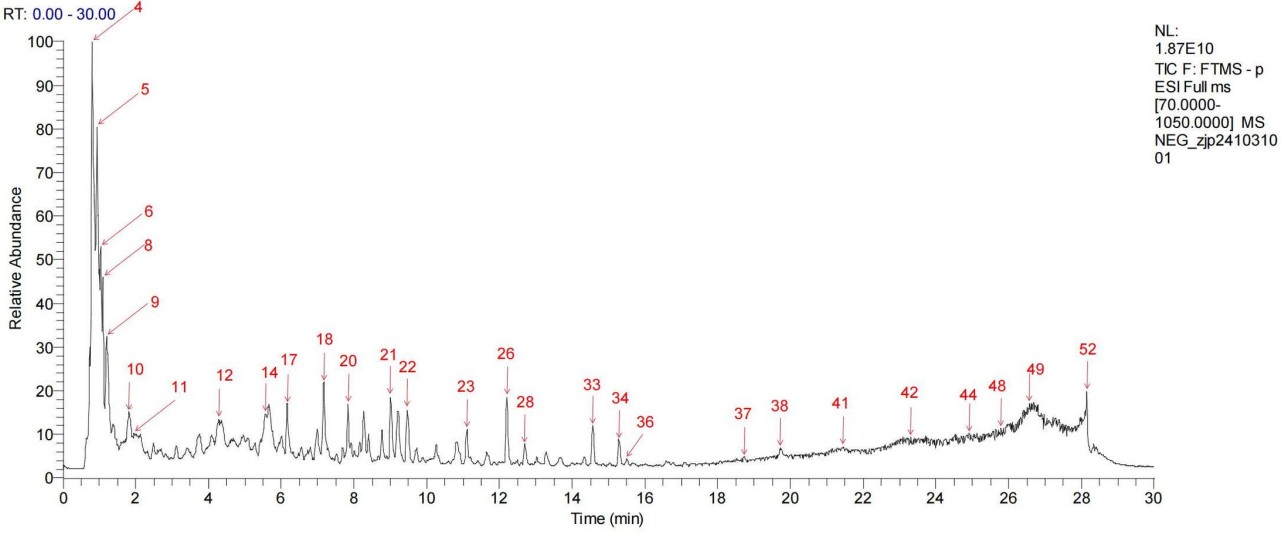

**Fig 1. Analyses of chemical constituents of HLWDD using UPLC-Q-TOF-MS.** Base peak chromatogram of HLWDD in the negative-ion MS mode.

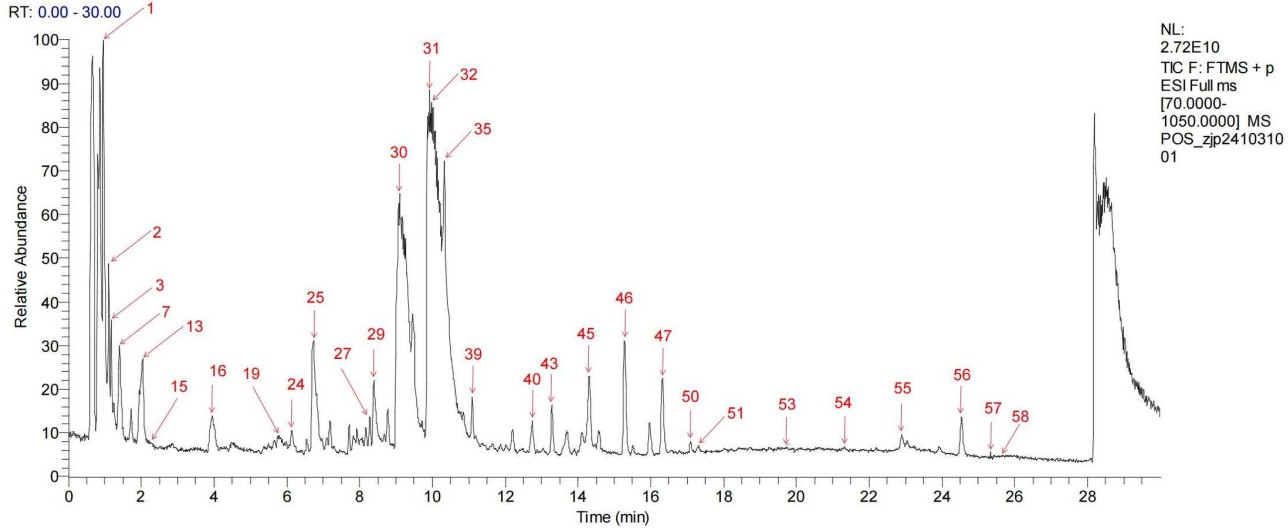

**Fig 2. Analyses of chemical constituents of HLWDD using UPLC-Q-TOF-MS.** Base peak chromatogram of HLWDD in the positive-ion MS mode.

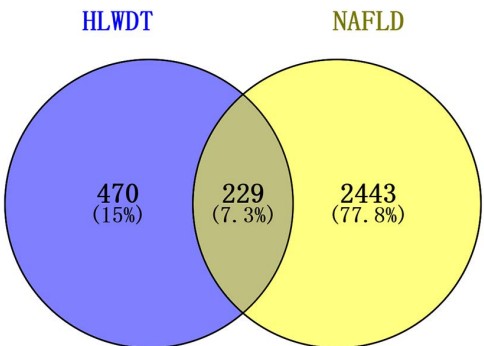

**Fig 3. Network pharmacology analysis of HLWDD against NAFLD.** Diagram of the target intersection.

Cytoscape 3.9.0. The top 10 hub targets based on degree values were TP53, AKT1, HSP90AA1, STAT3, EGFR, JUN, MTOR, CASP3, HSP90AB1, and MAPK1 (Fig 4).

GO and KEGG enrichment analyses via DAVID 6.8 revealed 363 GO terms: 265 biological processes (BP), 45 cellular components (CC), and 72 molecular functions (MF). Top BP terms included response to cadmium ion, regulation of gene expression and nitric oxide production, miRNA transcription, and inhibition of apoptosis (Fig 5). Dominant CC terms were cytoplasm, perinuclear region, protein complex, mitochondrion, and nucleus. Key MF terms included enzyme binding, protein phosphatase binding, and ubiquitin ligase interaction.

KEGG analysis yielded 148 enriched pathways; the top 20 are shown in Fig 6, including "Lipid and atherosclerosis," "Pathways in cancer," and "Fluid shear stress and atherosclerosis."

The compound-target network (Fig 7) highlighted the top 10 HLWDD components—Obacunone, Caffeic acid, Coptisine, Quercetin, Palmatine, Naringenin, Berberine, Isomeranzin, Tangeretin, and Epiberberine—based on pharmacological relevance and network centrality, along with corresponding targets such as EGFR, AKT1, MAPK1, F2, MAPK8, CYP1A2, CASP3, TNF, TP53, and STAT3.

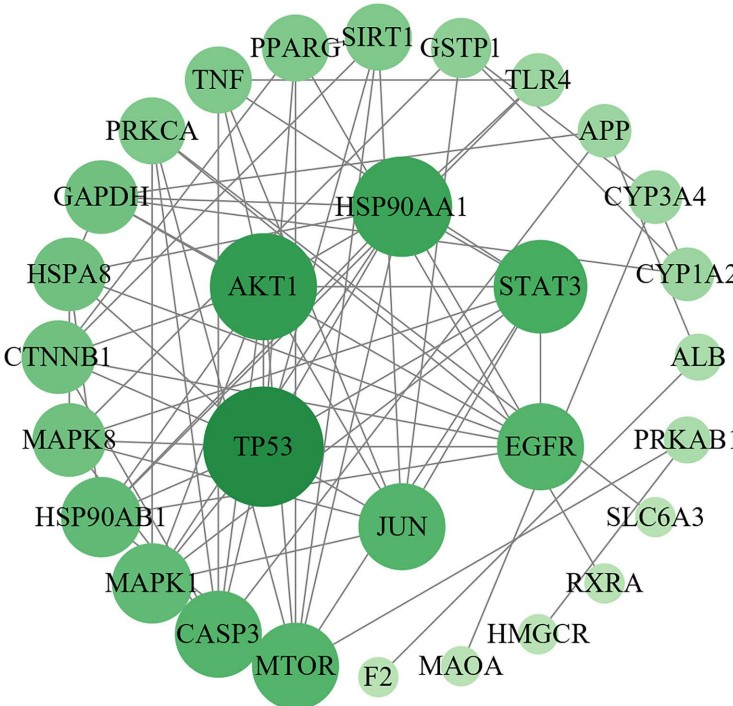

**Fig 4. Network pharmacology analysis of HLWDD against NAFLD.** Interaction network diagram of core target proteins of HLWDD.

We validated compound-target interactions via semi-flexible molecular docking using AutoDock. From the drug-ingredient-target-pathway network, the top 10 core compounds and 4 NAFLD-related targets (LKB1, AMPK, PPARα, CPT1A) were selected (Table 2). Binding energy < −5.0 kcal/mol indicated stable interactions; 8 of 24 pairs showed strong affinity (<−7.0 kcal/mol) (Fig 8), suggesting HLWDD compounds may modulate these targets to exert anti-NAFLD effects.

## Method validation and component quantification of HLWDD

Seven core HLWDD components were quantified based on HPLC analysis (Figs 9–11). All showed excellent linearity (r² ≥ 0.9993). For instance, quercetin (11.00–110.0 µg/mL, Y = 14,647X - 55,556) and berberine (13.00–130.0 µg/mL, Y = 2,046.2X - 8,890.9) displayed strong correlations. Method validation showed precision (RSD ≤ 1.25%), repeatability (RSD ≤ 2.20%), 24 h stability (RSD ≤ 1.62%), and high recovery (99.35–101.33%, RSD ≤ 1.53%). Epiberberine had the highest concentration (164.37 µg/mL), followed by obacunone (122.12 µg/mL) and naringenin (87.57 µg/mL) (Table 3). Detailed validation data are in S3 Table.

## Animal experiment verification

**Therapeutic effects of HLWDD on NAFLD rats.** As NAFLD progression is closely associated with abnormal lipid metabolism, serum biomarkers were assessed. Compared with the control group, the model group exhibited significantly elevated levels of TG, TC, and LDL-C (all $P < 0.01$), along with a significant reduction in HDL-C ($P < 0.01$). Treatment with high-dose HLWDD significantly decreased TG (P < 0.05), TC (P < 0.01), and LDL-C (P < 0.01), and significantly increased HDL-C levels (P < 0.01) (Figs 12–17).

Histopathological evaluation using H&E staining revealed clear differences among groups (Fig 18). The control group displayed normal hepatic architecture, characterized by polygonal hepatocytes with homogeneous eosinophilic cytoplasm,

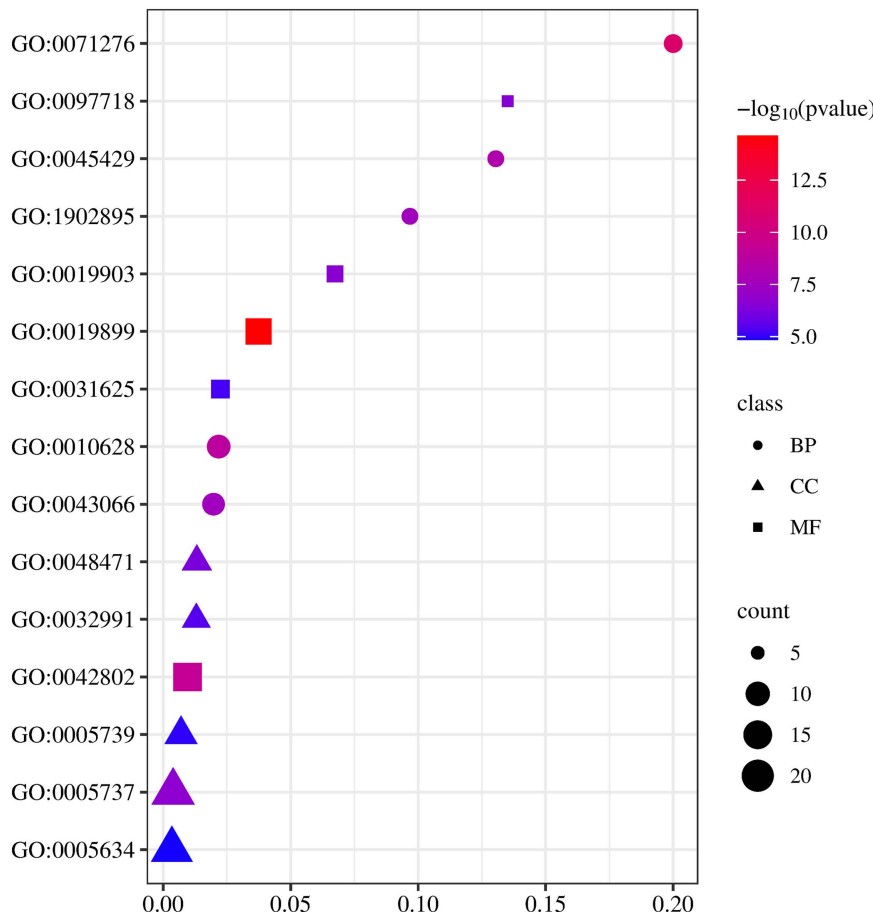

**Fig 5. Network pharmacology analysis of HLWDD against NAFLD.** KEGG pathway enrichment results pathway diagram.

centrally located nuclei, and intact lobular structure. In contrast, the model group exhibited classical features of NAFLD, including extensive cytoplasmic lipid vacuolization, eccentric nuclei, disorganized hepatic cords, and marked perivenular fat accumulation, confirming successful model establishment after 20 weeks of a high-fat/high-sugar diet.

As a clinically approved first-line anti-NAFLD agent, Metformin was included as a positive control. Compared with the control group, rats in the model group had significantly elevated serum IL-6, IL-1β, and TNF-α levels (all $P<0.001$) (Figs 19–21). These inflammatory markers were significantly reduced following high-dose HLWDD and Metformin treatment, indicating potential anti-inflammatory effects.

**Effects of HLWDD on AMPK/PPARα signaling.** To investigate the regulatory effect of HLWDD on lipid metabolism signaling, we assessed both protein and gene expression of the AMPK/PPARα pathway in liver tissue (Figs 22–30).

Western blot analysis showed that the expression of AMPK, CPT1A, LKB1, and PPARα proteins was significantly decreased in the model group compared to the control ($P<0.05$). High-dose HLWDD treatment significantly restored protein expression of all four markers ($P<0.05$) (Figs 22–26).

Quantitative PCR results similarly indicated significant downregulation of Ampk, Cpt1a, Lkb1, and Ppara mRNA levels in the model group ($P<0.05$), which were significantly upregulated following HLWDD administration ($P<0.05$) (Figs 27–30).

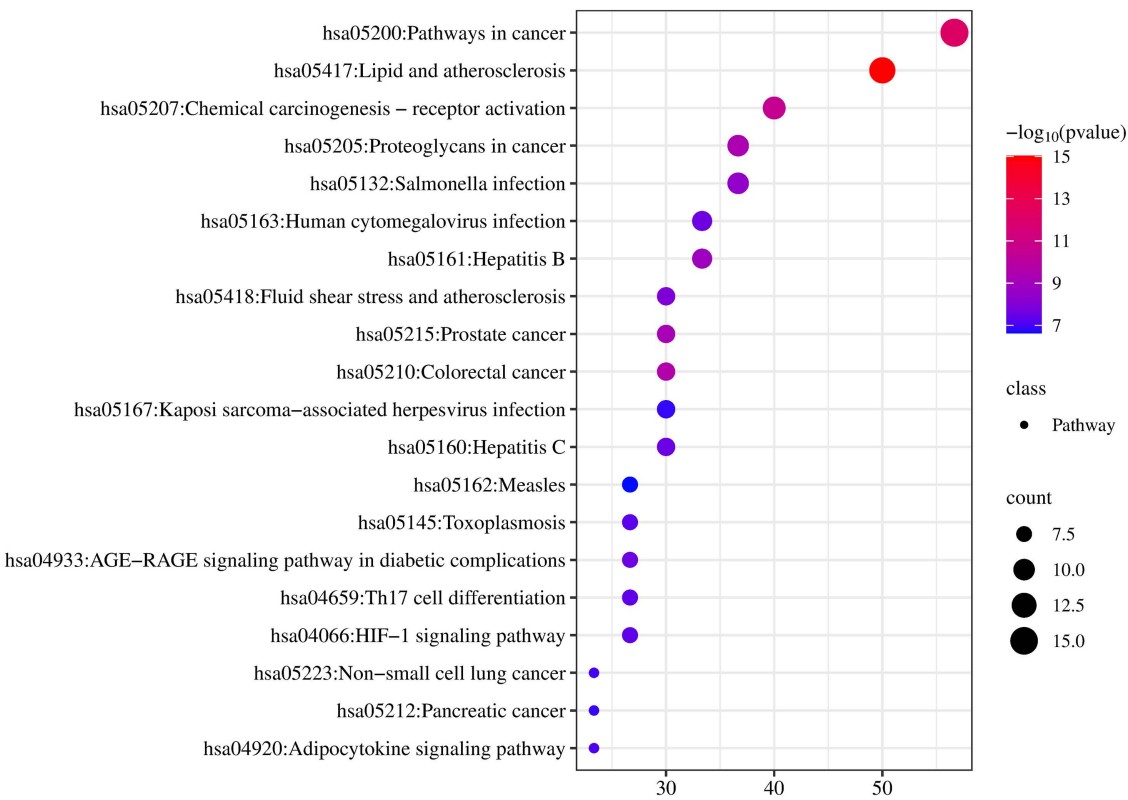

**Fig 6. Network pharmacology analysis of HLWDD against NAFLD.** GO enrichment result bubble chart.

**Multi-omics integration: Gut-liver axis and AMPK activation.** Metagenomics and metabolomics identified 13 differential metabolites, including fatty acid derivatives (e.g., N-palmitoylthreonine, sphingosine), amino acids (lysine, isoleucine, aspartic acid), and bile acid-related metabolites (e.g., enterolactone) (Fig 31).

*Akkermansia* positively correlated with aspartic acid ($r = 0.71$) and pantothenic acid ($r = 0.65$) (Fig 32). It may alleviate NAFLD by: (1) activating AMPK and promoting PPARα-mediated β-oxidation, and (2) enhancing aspartate metabolism to reduce oxidative stress [22].

Model rats showed disrupted microbiota-metabolite correlations (e.g., with allysine, L-isoleucine), which HLWDD restored. HLWDD also increased N-palmitoylthreonine (+58%, $P < 0.01$) and enterolactone (+43%, $P < 0.01$) (Fig 33), supporting AMPK-mediated lipid regulation.

Pantothenic acid levels decreased by 42% in model rats ($P < 0.01$), but increased by 37% after HLWDD-H treatment ($P < 0.01$) (Fig 34).

Heatmap analysis showed strong correlations between key proteins (LKB1, AMPK, CPT1A) and metabolites (e.g., aspartic acid: $r = 0.82$; pantothenic acid: $r = 0.76$), confirming coordinated regulation of lipid oxidation and providing mechanistic insight (Fig 35).

## Discussion

NAFLD, a common chronic liver condition, is strongly associated with insulin resistance, inflammation, and hepatic lipid metabolic dysfunction [23]. Recent studies have suggested that intestinal flora contributes to NAFLD development by influencing inflammation, oxidative stress, and hepatic energy metabolism through microbial metabolites [23]. Traditional

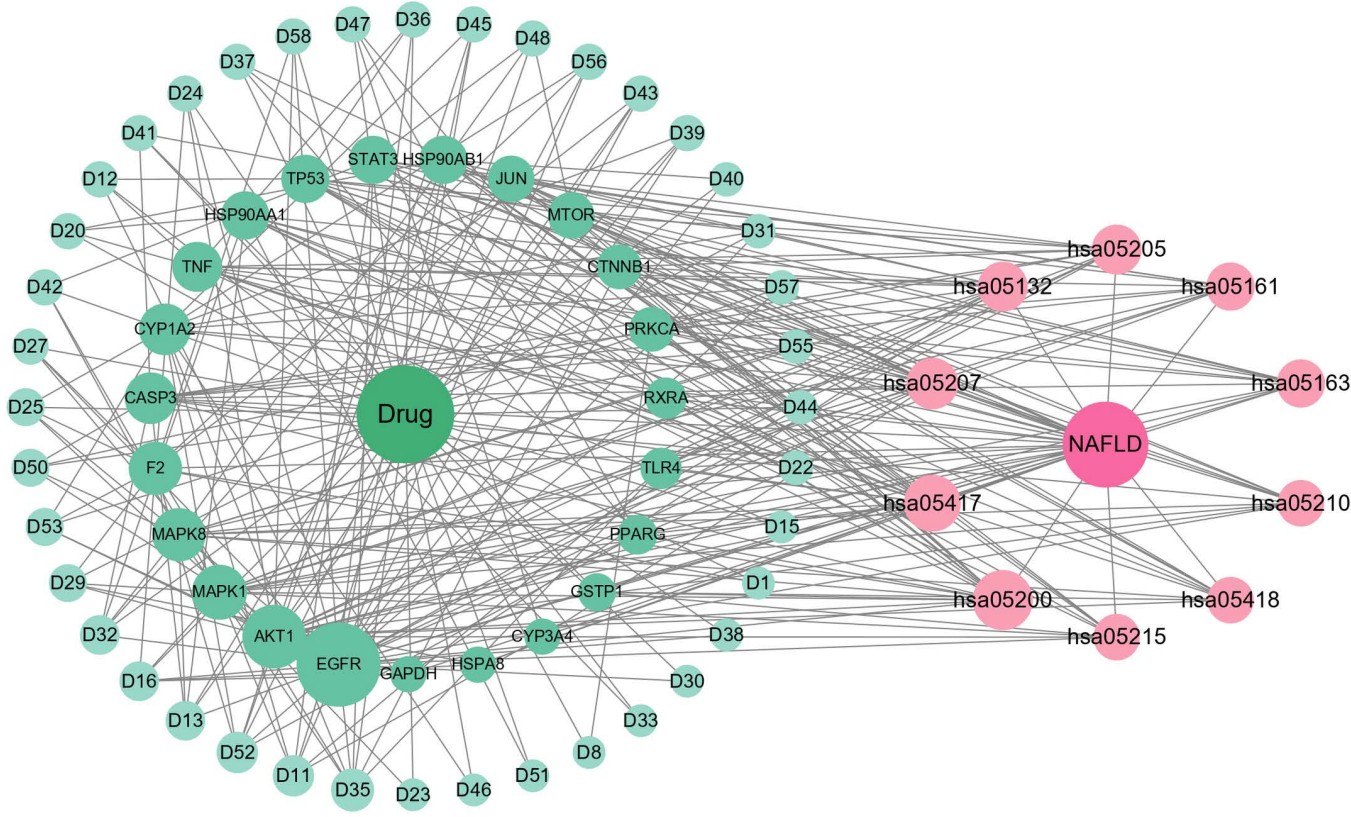

**Fig 7. Network pharmacology analysis of HLWDD against NAFLD.** Drug–ingredients–genes–pathway interaction network diagram.

**Table 2. Binding energy of core components and key core targets in HLWDD.**

| NO. | Compoud | Binding Energy/kcal·mol⁻¹ | | | |
|---|---|---|---|---|---|
| | | LKB1 | AMPK | PPARα | CPT1A |
| 1 | Coptisine | −5.79 | −6.59 | −8.15 | −7.35 |
| 2 | Phellodendrine | −4.63 | −5.45 | −7.36 | −5.24 |
| 3 | Berberine | −5.42 | −7.08 | −6.39 | −4.32 |
| 4 | Epiberberine | −5.28 | −4.24 | −7.32 | −6.32 |
| 5 | Quercetin | −7.66 | −6.25 | −5.23 | −4.11 |
| 6 | Palmatine | −7.55 | −5.52 | −5.36 | −6.48 |
| 7 | Naringenin | −5.14 | −6.32 | −5.01 | −8.17 |

Chinese medicine has garnered interest for its potential to prevent and treat NAFLD. HLWDD, a classical formulation with reputed effects of clearing heat and phlegm, soothing the stomach, and promoting bile secretion, may exert protective effects in NAFLD. This study explored whether HLWDD may modulate intestinal microbiota and metabolic profiles, potentially activating the LKB1/AMPK pathway to attenuate NAFLD progression. Given that AMPK activation is clinically associated with attenuated hepatocellular injury (e.g., reduced ALT/AST) and enhanced insulin sensitivity [24], our molecular findings imply HLWDD may confer similar hepatoprotective and metabolic benefits via AMPK activation. Specifically, the LKB1/AMPK pathway plays a central role in maintaining hepatic energy homeostasis, inhibiting lipogenesis, and enhancing insulin sensitivity [25]. AMPK activation can suppress lipogenic genes such as SREBP-1c and FAS, reducing hepatic

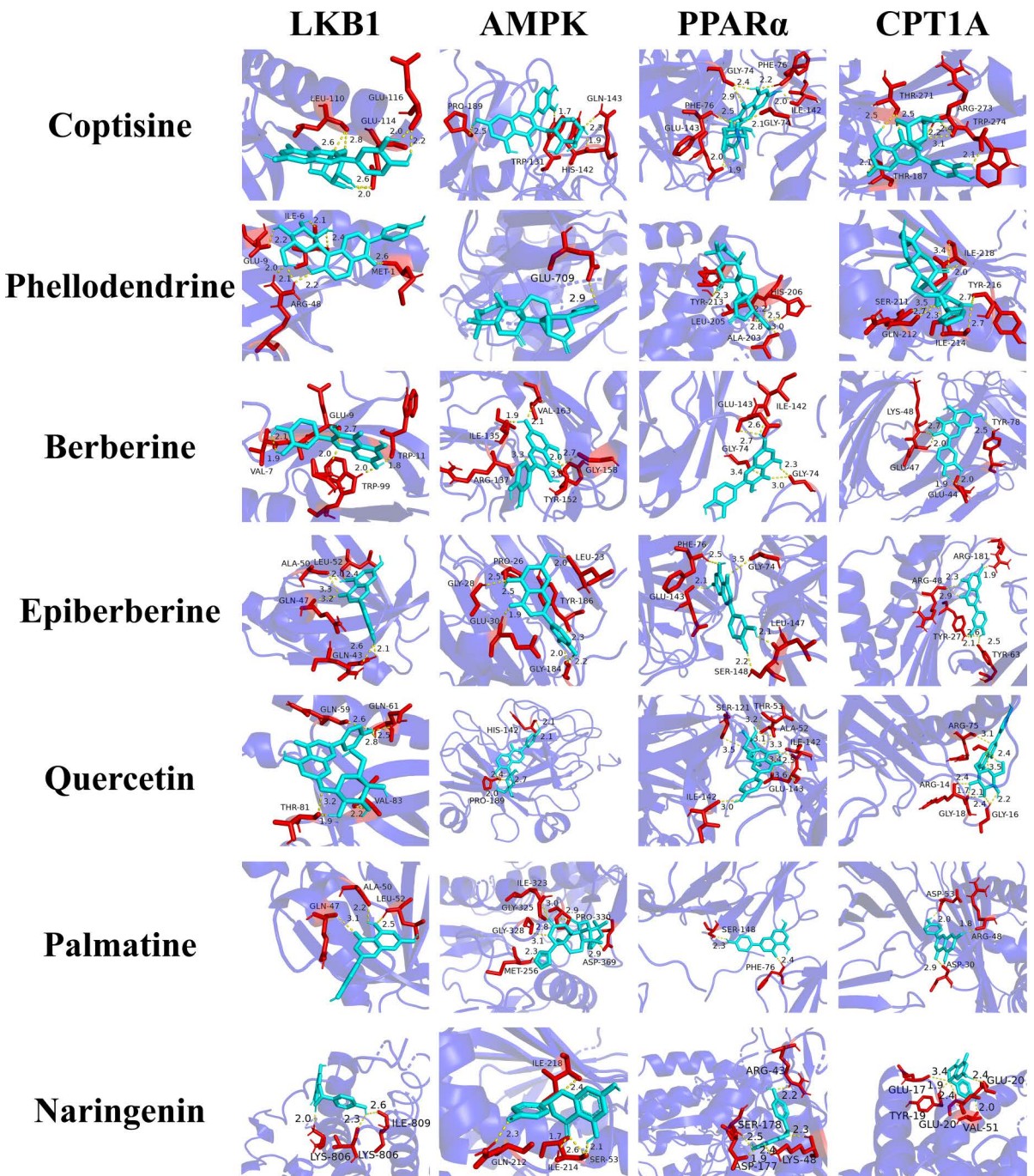

**Fig 8. Molecular docking validation of HLWDD against NAFLD.** Representative docking results.

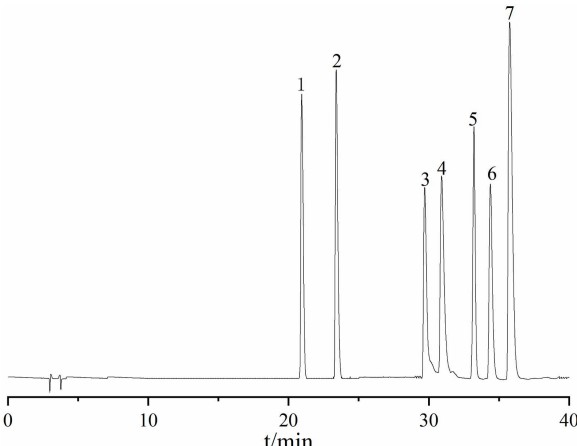

**Fig 9. HPLC chromatograms of HLWDD.** Mixed reference substance. Peaks: 1. Quercetin, 2. Epiberberine, 3. Coptisine, 4. Palmatine, 5. Berberine, 6. Naringenin, 7. Houttuynia cordata.

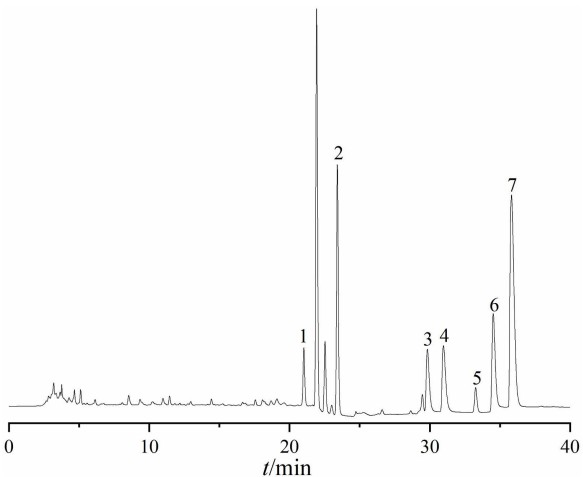

**Fig 10. HPLC chromatograms of HLWDD.** Sample extract. Peaks: 1. Quercetin, 2. Epiberberine, 3. Coptisine, 4. Palmatine, 5. Berberine, 6. Naringenin, 7. Houttuynia cordata.

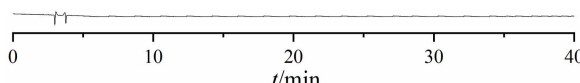

**Fig 11. HPLC chromatograms of HLWDD.** Blank control. Peaks: 1. Quercetin, 2. Epiberberine, 3. Coptisine, 4. Palmatine, 5. Berberine, 6. Naringenin, 7. Houttuynia cordata.

fat accumulation [26]. In this study, HLWDD intervention increased hepatic LKB1 and p-AMPK protein expression while reducing SREBP-1c and FAS levels. Mechanistically, AMPK activation suppresses lipogenesis and promotes GLUT4 translocation while inhibiting PGC-1α-mediated gluconeogenesis [27]—collectively suggesting improved hepatic glucose/lipid homeostasis.

**Table 3. Contents of seven ingredients in HLWDD.**

| Compound | Content (µg/mL) | | | | | | |
|---|---|---|---|---|---|---|---|
| | Quercetin | Epiberberine | Coptisine | Palmatine | Berberine | Naringenin | Obacunone |
| 1 | 15.47 | 162.34 | 35.22 | 30.24 | 15.78 | 87.14 | 121.25 |
| 2 | 14.99 | 165.79 | 35.12 | 31.09 | 15.67 | 87.90 | 122.76 |
| 3 | 14.87 | 164.99 | 34.76 | 30.79 | 15.55 | 87.66 | 122.34 |
| Average value | 15.11 | 164.37 | 35.03 | 30.71 | 15.67 | 87.57 | 122.12 |
| RSD(%) | 0.32 | 1.81 | 0.24 | 0.43 | 0.12 | 0.39 | 0.78 |

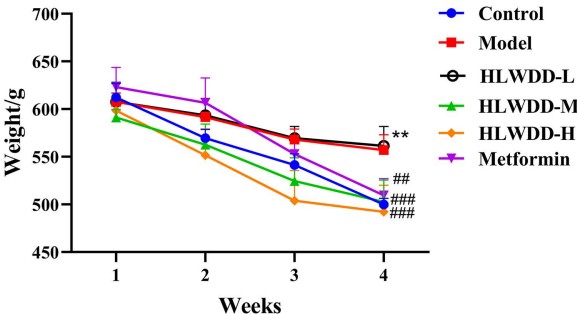

**Fig 12. Therapeutic effects of HLWDD on NAFLD rats.** Body weight changes during intervention. Data expressed as mean ± SEM (n = 3). Statistical significance: **$P < 0.01$, ***$P < 0.001$ vs. Control; #$P < 0.05$, ##$P < 0.01$, ###$P < 0.001$ vs. Model.

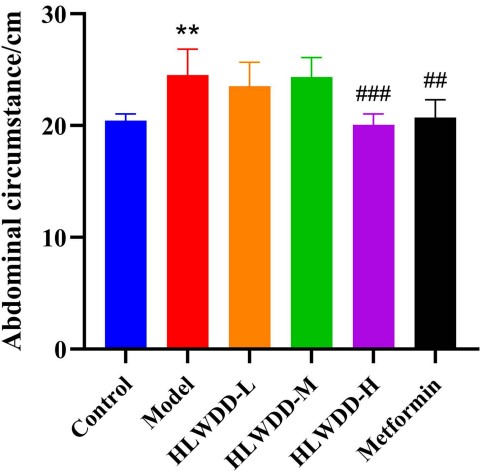

**Fig 13. Therapeutic effects of HLWDD on NAFLD rats.** Abdominal circumference changes during intervention. Data expressed as mean ± SEM (n = 3). Statistical significance: same as Fig 12.

Given HLWDD's modulation of gut microbiota, we focused on SCFAs—key metabolites produced by intestinal flora—as important mediators of host energy metabolism [28]. Our results showed that HLWDD treatment increased the abundance of *Akkermansia* muciniphila, a mucus-degrading bacterium associated with improved gut barrier integrity and metabolic regulation [29]. Previous research indicates that *Akkermansia* colonization correlates with amino acid metabolism, particularly aspartate, which may contribute to hepatic energy balance through the gut-liver amino acid axis [30]. SCFAs such as butyrate are known to regulate lipid metabolism via GPR43/41 and AMPK signaling pathways [31].

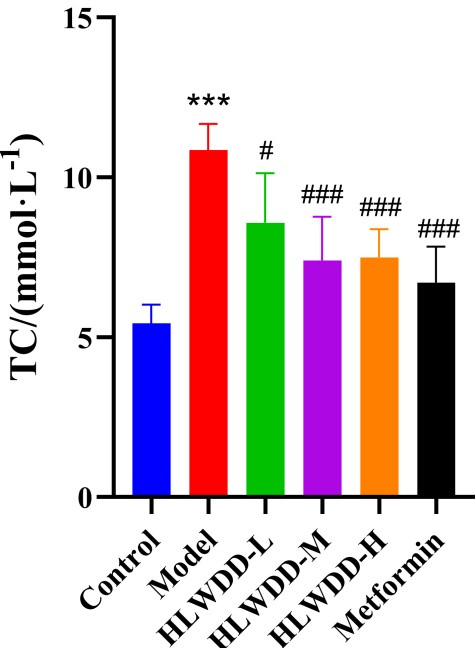

**Fig 14. Therapeutic effects of HLWDD on NAFLD rats.** Serum TC levels. Data expressed as mean ± SEM (n = 3). Statistical significance: same as Fig 12.

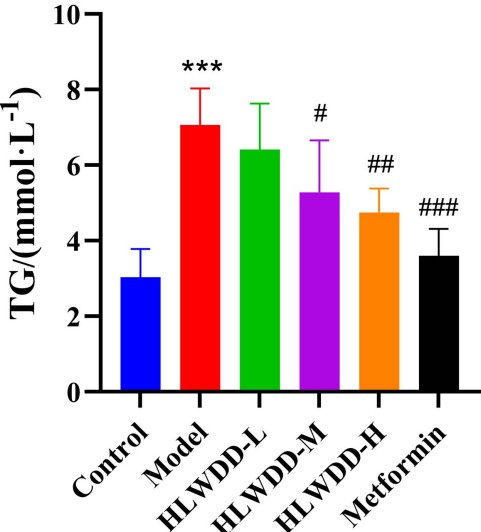

**Fig 15. Therapeutic effects of HLWDD on NAFLD rats.** Serum TG levels. Data expressed as mean ± SEM (n = 3). Statistical significance: same as Fig 12.

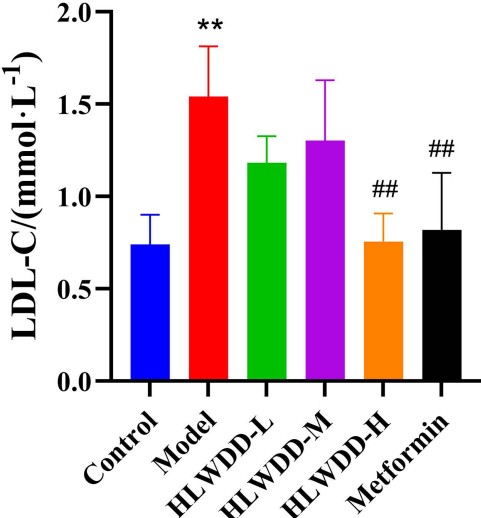

**Fig 16. Therapeutic effects of HLWDD on NAFLD rats.** Serum LDL-C levels. Data expressed as mean±SEM (n=3). Statistical significance: same as Fig 12.

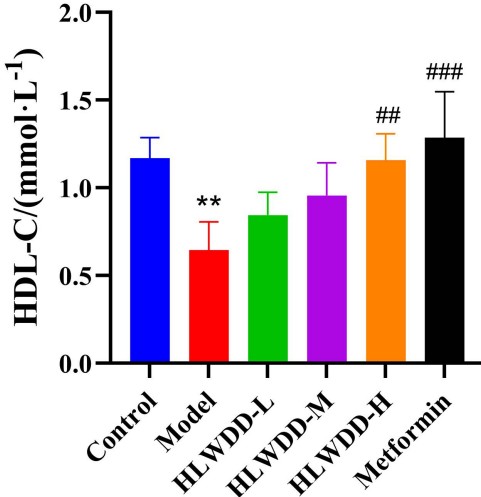

**Fig 17. Therapeutic effects of HLWDD on NAFLD rats.** Serum HDL-C levels. Data expressed as mean±SEM (n=3). Statistical significance: same as Fig 12.

This study suggests that HLWDD may modulate the gut microbiota in NAFLD rats, increase butyrate-producing bacteria, enhance SCFA levels, and help restore bile acid homeostasis. Metabolomics analysis further revealed increased serum and hepatic aspartate levels following HLWDD treatment. Literature reports support that aspartate may enhance mitochondrial function and suppress lipid synthesis by activating the LKB1/AMPK pathway [32]. Thus, HLWDD may facilitate a regulatory network involving microbiota, metabolites, and signaling pathways.

It is noteworthy that the relationship between aspartate metabolism and LKB1/AMPK activation appears to be bidirectional. AMPK activation can promote aspartate transporter expression, increasing aspartate availability in hepatocytes [33]. Conversely, aspartate, a mitochondrial intermediate, may also modulate LKB1 phosphorylation activity [34].

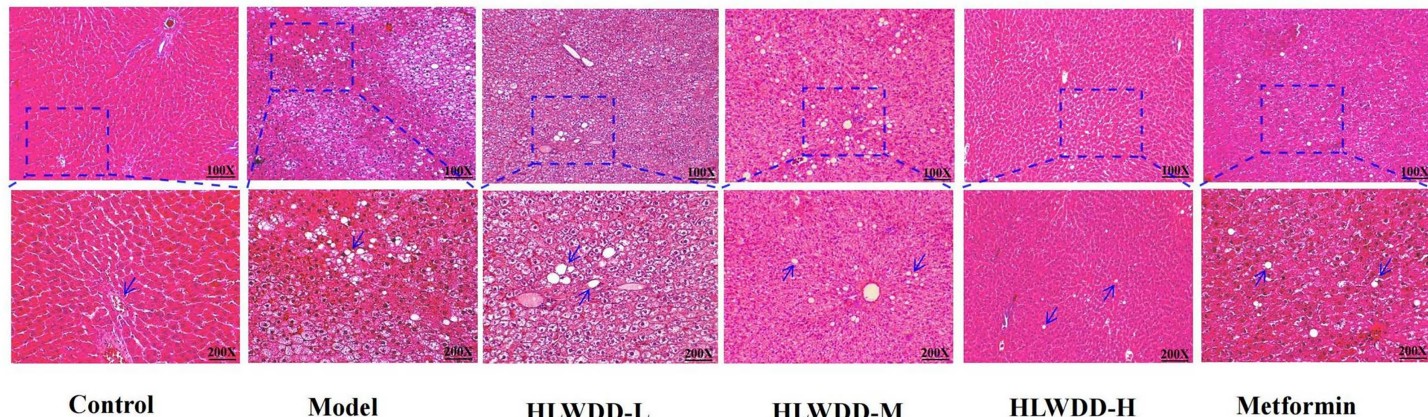

| Control | Model | HLWDD-L | HLWDD-M | HLWDD-H | Metformin |

**Fig 18. Therapeutic effects of HLWDD on NAFLD rats.** Representative H&E staining of liver tissues (scale bar = 100 µm).

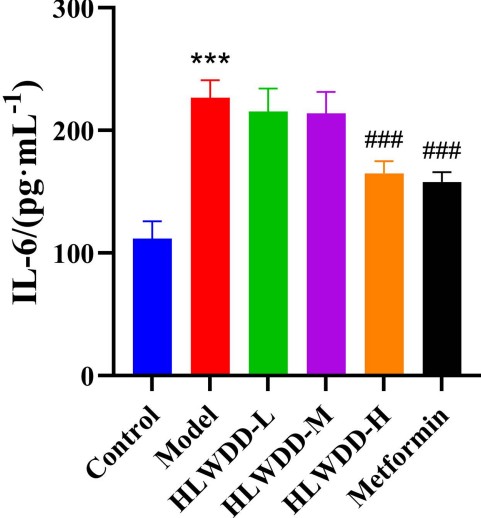

**Fig 19. Therapeutic effects of HLWDD on NAFLD rats.** Serum IL-6 levels. Data expressed as mean ± SEM (n = 3). Statistical significance: same as Fig 12.

Correlation analysis in this study showed a strong association between hepatic aspartate levels and the p-AMPK/AMPK ratio, as well as between *Akkermansia* abundance and liver aspartate concentration. This suggests that HLWDD establishes a self-reinforcing cycle that sustains metabolic improvements by enhancing *Akkermansia*-associated aspartate flux and LKB1/AMPK activity.

Beyond these metabolic benefits, inflammation is a major contributor to NAFLD progression. Liver-resident Kupffer cells, when overactivated, release inflammatory cytokines such as TNF-α and IL-6 via the NF-κB pathway, exacerbating hepatic injury [35]. Our findings showed that HLWDD reduced the phosphorylation of NF-κB p65 and downregulated these inflammatory cytokines. Critically, combined with microbiota data indicating enhanced gut barrier integrity and SCFA production, these results suggest HLWDD attenuates hepatic inflammation at least partly by modulating gut-derived metabolites—a mechanism unaddressed by conventional NAFLD pharmacotherapies. Consequently, HLWDD demonstrates superior multi-targeted efficacy compared to existing therapies: Metformin only partially activates AMPK while failing to correct gut

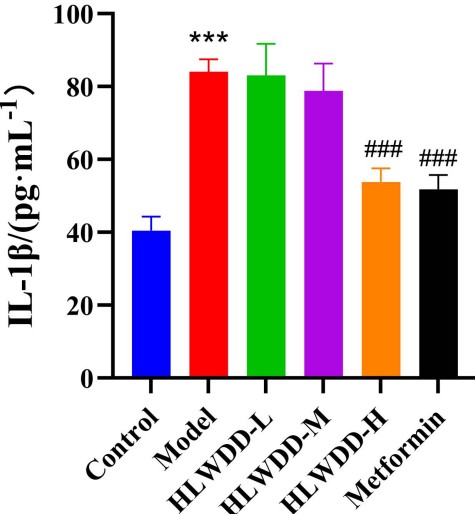

**Fig 20. Therapeutic effects of HLWDD on NAFLD rats.** Serum IL-1β levels. Data expressed as mean±SEM (n=3). Statistical significance: same as Fig 12.

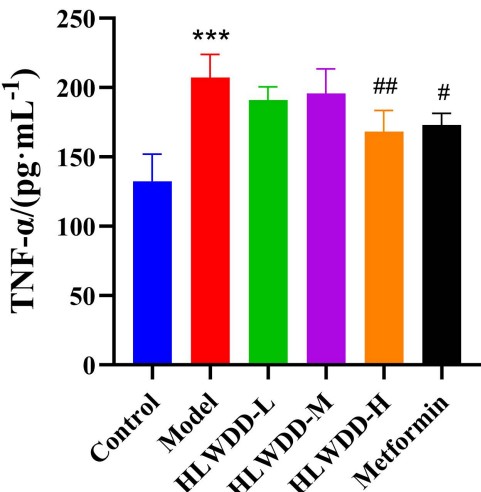

**Fig 21. Therapeutic effects of HLWDD on NAFLD rats.** Serum TNF-α levels. Data expressed as mean±SEM (n=3). Statistical significance: same as Fig 12.

dysbiosis [36]; pioglitazone significantly increases cardiovascular risk [37]; whereas HLWDD concurrently resolves inflammation through gut-liver axis modulation while avoiding safety concerns associated with conventional agents.

Despite the promising findings, several limitations of this study must be acknowledged. First, only one animal model was used, which limits generalizability. Second, the study did not include a dose-response analysis or compare HLWDD with established therapies beyond metformin. Third, while associations between HLWDD, the microbiota, and metabolic pathways were observed, no functional validation—such as AMPK inhibition or germ-free animal experiments—was performed to confirm causality. Future studies should address these limitations by including multiple models, conducting pharmacodynamic comparisons, and applying targeted interventions to verify mechanisms.

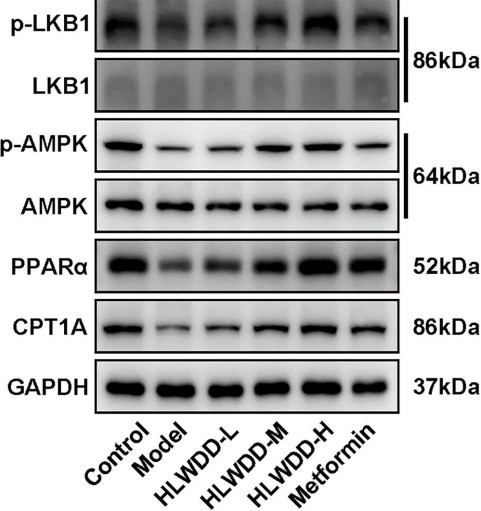

**Fig 22. Effects of HLWDD on AMPK/PPARα signaling in rat liver.** Western blot bands of p-LKB1, LKB1, p-AMPK, AMPK, PPARα, CPT1A, and GAPDH.

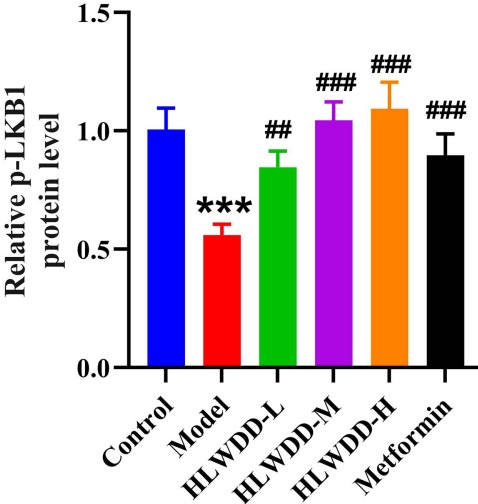

**Fig 23. Effects of HLWDD on AMPK/PPARα signaling in rat liver.** Relative protein expression of p-LKB1. Data are statistically presented as means ± SEM (n = 3). Compared with the Control, **$P < 0.01$, ***$P < 0.001$; compared with the Model, #$P < 0.05$, ##$P < 0.01$, ###$P < 0.001$.

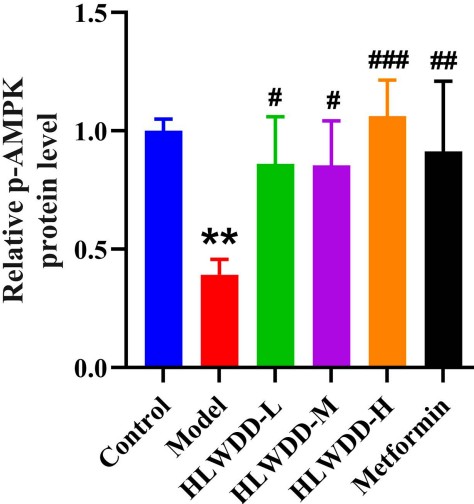

**Fig 24. Effects of HLWDD on AMPK/PPARα signaling in rat liver.** Relative protein expression of p-AMPK. Data expressed as mean±SEM (n=3). Statistical significance: same as Fig 23.

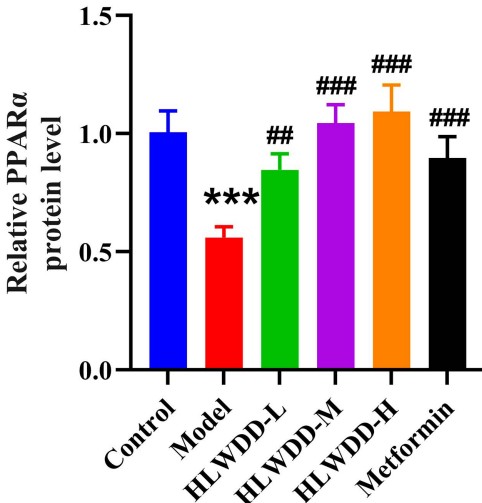

**Fig 25. Effects of HLWDD on AMPK/PPARα signaling in rat liver.** Relative protein expression of PPARα. Data expressed as mean±SEM (n=3). Statistical significance: same as Fig 23.

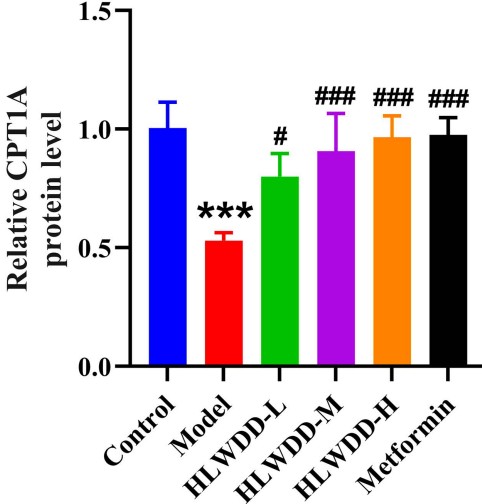

**Fig 26. Effects of HLWDD on AMPK/PPARα signaling in rat liver.** Relative protein expression of CPT1A. Data expressed as mean ± SEM (n = 3). Statistical significance: same as Fig 23.

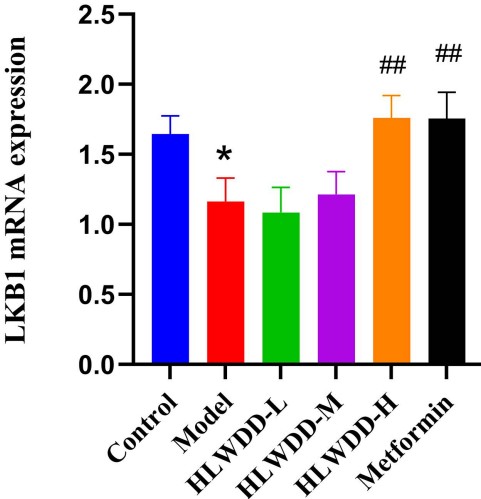

**Fig 27. Effects of HLWDD on AMPK/PPARα signaling in rat liver.** Relative mRNA expression of LKB1. Data expressed as mean ± SEM (n = 3). Statistical significance: same as Fig 23.

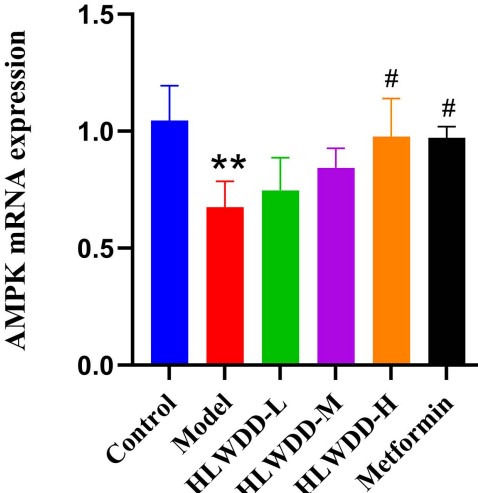

**Fig 28. Effects of HLWDD on AMPK/PPARα signaling in rat liver.** Relative mRNA expression of AMPK. Data expressed as mean ± SEM (n = 3). Statistical significance: same as Fig 23.

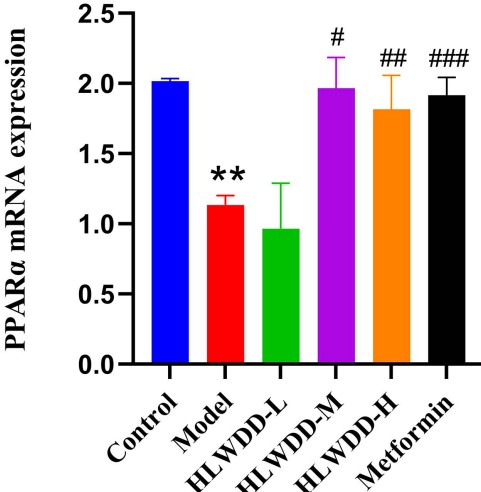

**Fig 29. Effects of HLWDD on AMPK/PPARα signaling in rat liver.** Relative mRNA expression of PPARα. Data expressed as mean ± SEM (n = 3). Statistical significance: same as Fig 23.

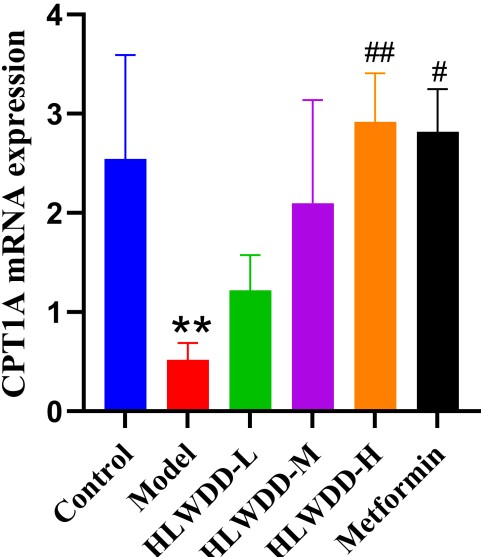

**Fig 30. Effects of HLWDD on AMPK/PPARα signaling in rat liver.** Relative mRNA expression of CPT1A. Data expressed as mean±SEM (n=3). Statistical significance: same as Fig 23.

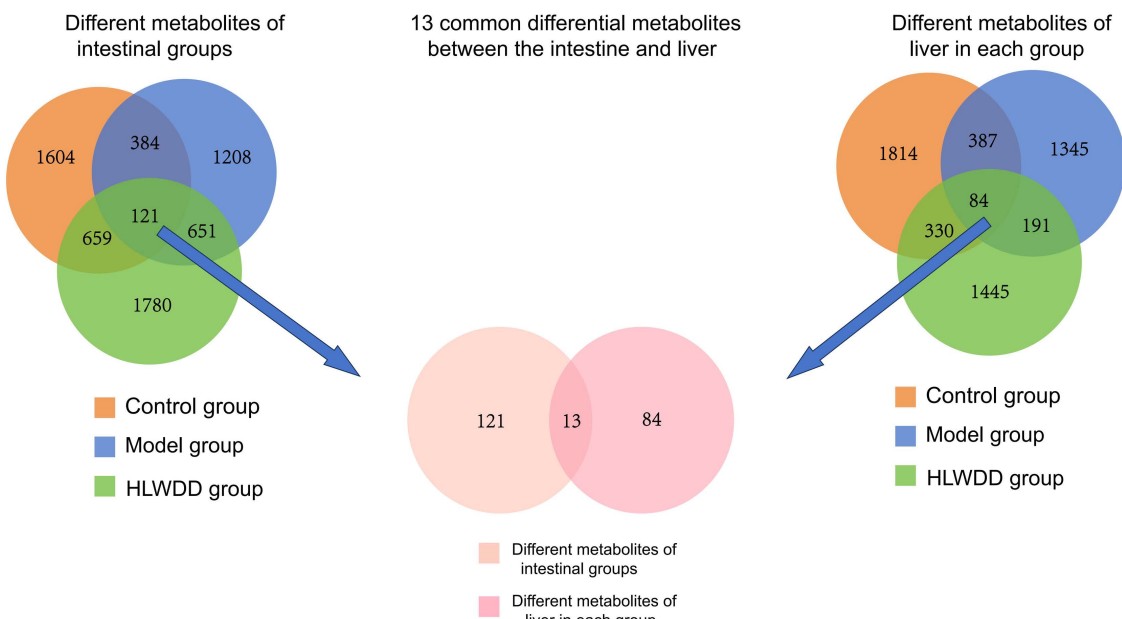

**Fig 31. Multi-omics analysis of HLWDD-mediated gut–liver axis regulation in NAFLD.** Venn diagram of differential metabolites identified by gut metagenomics and liver metabolomics.

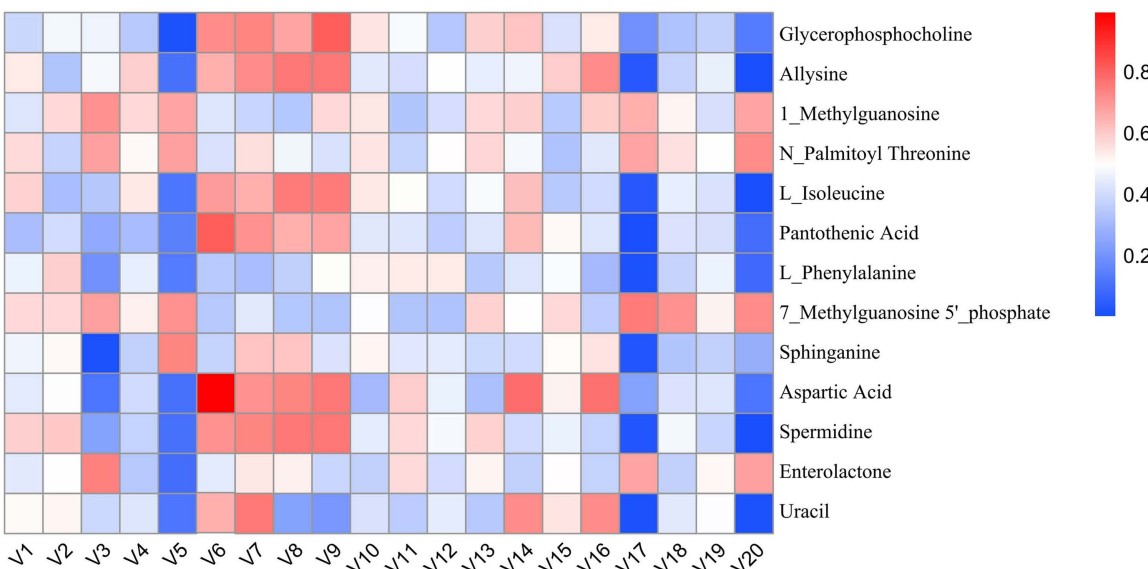

**Fig 32. Multi-omics analysis of HLWDD-mediated gut–liver axis regulation in NAFLD.** Heatmap showing correlations between *Akkermansia* and key metabolites (aspartic acid, pantothenic acid).Color scales indicate Pearson correlation coefficients (r). Arrows highlight HLWDD-induced restorative trends. Data expressed as mean±SEM (n=3). Statistical significance: *P*<0.05 (red), *P*<0.01 (dark red).

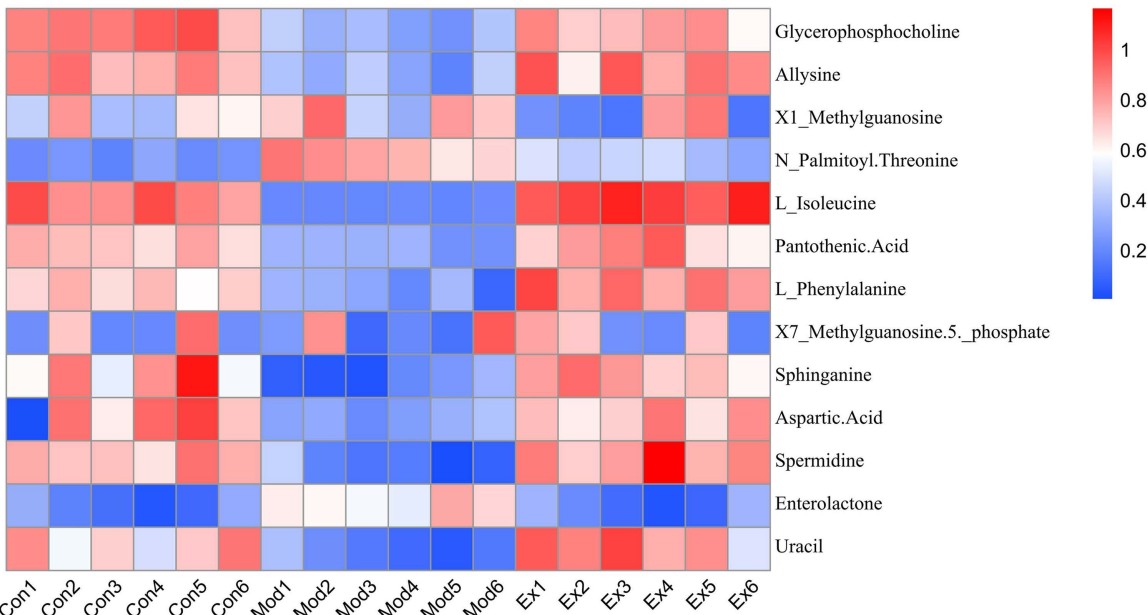

**Fig 33. Multi-omics analysis of HLWDD-mediated gut–liver axis regulation in NAFLD.** Gut microbiota–metabolite correlation network in model vs. HLWDD groups. Statistical significance: same as Fig 32.

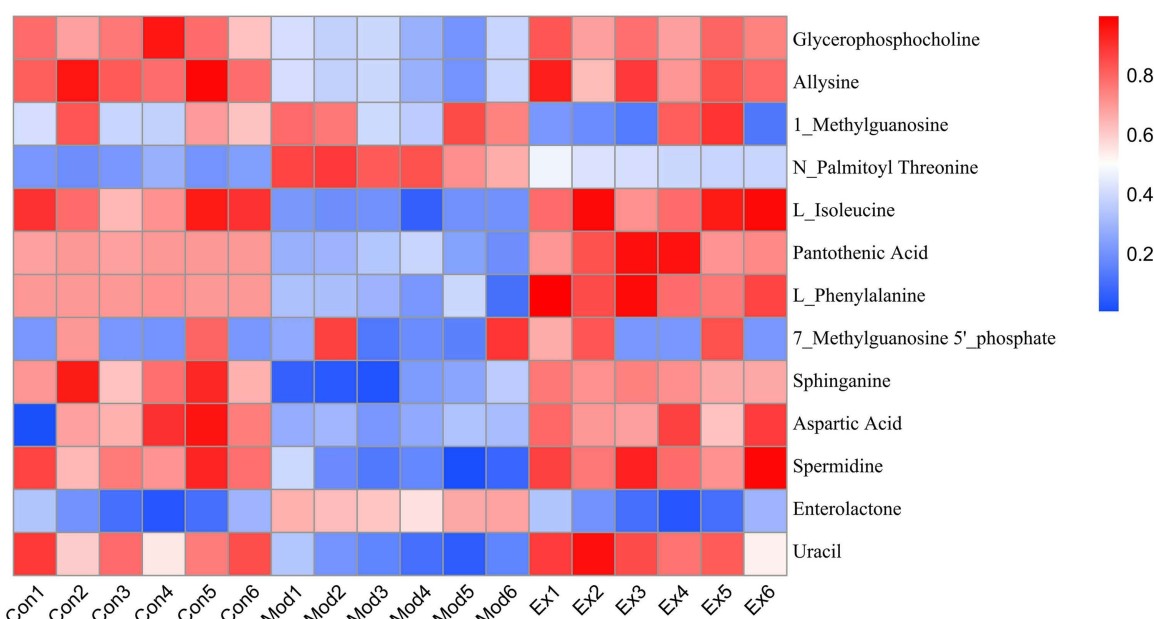

**Fig 34. Multi-omics analysis of HLWDD-mediated gut–liver axis regulation in NAFLD.** Hepatic metabolite correlation shifts (model vs. HLWDD-H). Statistical significance: same as Fig 32.

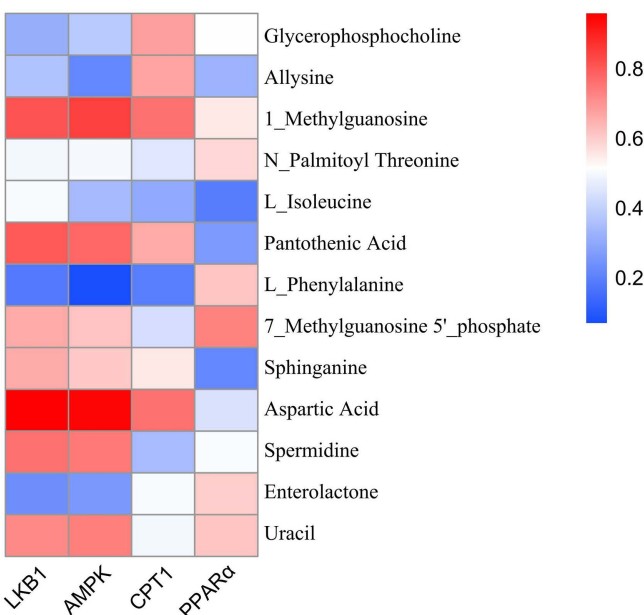

**Fig 35. Multi-omics analysis of HLWDD-mediated gut–liver axis regulation in NAFLD.** Integrated heatmap of metabolite–protein interactions (LKB1/AMPK/CPT1A). Statistical significance: same as Fig 32.

In conclusion, this study suggests that HLWDD may alleviate NAFLD by modulating gut microbiota and associated metabolites and by activating the LKB1/AMPK pathway. These findings provide experimental support for HLWDD's therapeutic potential and underscore the role of the gut-liver axis in NAFLD pathogenesis.

## Supporting information

**S1 Table. Chemical constituents identified in HLWDD by UPLC-Q-TOF-MS. Retention times, mass spectral data, and compound classifications.**
(DOCX)

**S2 Table. Shared targets between HLWDD active compounds and NAFLD.** Full list of 229 targets identified through network pharmacology.
(DOCX)

**S3 Table. Method validation data for quantification of seven HLWDD components.** Linear regression equations, precision, repeatability, stability, and recovery results.
(DOCX)

## Author contributions

**Conceptualization:** Jianping Zhu, Ji Li.

**Data curation:** Jianping Zhu, Yuzhen Chen, Ji Li.

**Formal analysis:** Yuzhen Chen, Yidi Han, Ji Li.

**Funding acquisition:** Ji Li.

**Methodology:** Jianping Zhu, Yuzhen Chen, Ji Li.

**Project administration:** Ji Li.

**Visualization:** Jianping Zhu, Ji Li.

**Writing – original draft:** Jianping Zhu, Ji Li.

**Writing – review & editing:** Jianping Zhu, Ji Li.

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
