## [Decision Letter · Decision Letter 0]

3 Jul 2025

PONE-D-25-23758

Mechanism of Huanglian Wendan Decoction in Ameliorating Non-Alcoholic Fatty Liver Disease via Modulating Gut Microbiota-Mediated Metabolic Reprogramming and Activating the LKB1/AMPK Pathway

PLOS ONE

Dear Dr. Li,

Thank you for submitting your manuscript to PLOS ONE. After careful consideration, we feel that it has merit but does not fully meet PLOS ONE’s publication criteria as it currently stands. Therefore, we invite you to submit a revised version of the manuscript that addresses the points raised during the review process.

Your manuscript has been reviewed by three experts in the field. My additional comment is as follows: Please provide the raw data related to the gut microbiome analysis conducted in this study. This is essential to validate the reliability and integrity of the analysis. Kindly ensure this data is included in your revised submission.

Request from the Editorial Office: In your Methods section, please provide the following information: 1) The source of the plants used in the study, their authentication, and which parts of each plant were used to produce the decoction. 2) How was the decoction given to the animals, and what is the relevance of the in vivo assays to its traditional use? 3) What is the rationale for the concentrations of the medicinal compound used in the experiments? We would expect an acute toxicity test to have been performed. 4) How were the animals modelled?

We look forward to receiving your revised manuscript.

Kind regards,

Fahrul Nurkolis

Academic Editor

PLOS ONE

Journal Requirements: 

4. To comply with PLOS ONE submissions requirements, in your Methods section, please provide additional information regarding the experiments involving animals and ensure you have included details on methods of sacrifice, and efforts to alleviate suffering.

6. Thank you for stating the following financial disclosure:

 [This work was supported by Hunan Provincial Department of Education Research Project (23B0375); Hunan Provincial Administration of Traditional Chinese Medicine Research Project (C2023021); Hunan University of Traditional Chinese Medicine University-level Research Project (2024XJZC017).]. 

7.  Thank you for stating the following in the Acknowledgments Section of your manuscript:

[This work was supported by Hunan Provincial Department of Education Research Project (23B0375); Hunan Provincial Administration of Traditional Chinese Medicine Research Project (C2023021); Hunan University of Traditional Chinese Medicine University-level Research Project (2024XJZC017).]

  [This work was supported by Hunan Provincial Department of Education Research Project (23B0375); Hunan Provincial Administration of Traditional Chinese Medicine Research Project (C2023021); Hunan University of Traditional Chinese Medicine University-level Research Project (2024XJZC017).]. 

8. Your ethics statement should only appear in the Methods section of your manuscript. If your ethics statement is written in any section besides the Methods, please delete it from any other section.

Reviewers' comments:

Reviewer's Responses to Questions

**Comments to the Author**

1. Is the manuscript technically sound, and do the data support the conclusions?

Reviewer #1: Yes

Reviewer #2: Yes

Reviewer #3: Yes

2. Has the statistical analysis been performed appropriately and rigorously? 

Reviewer #1: Yes

Reviewer #2: No

Reviewer #3: Yes

3. Have the authors made all data underlying the findings in their manuscript fully available?

Reviewer #1: Yes

Reviewer #2: No

Reviewer #3: Yes

4. Is the manuscript presented in an intelligible fashion and written in standard English?

Reviewer #1: Yes

Reviewer #2: Yes

Reviewer #3: Yes

5. Review Comments to the Author

Reviewer #1: 1. Abstract

Strengths : Clearly presents important topic, comprehensive methodology, mechanistic insight and strong outcome summary

Suggestions for improvement :

The abstract would benefit from clearer segmentation into logical components : Background, objective, methods, key results, conclusion. Research objective need explicit statement. Too many abbreviations may overwhelm readers unfamiliar with them. The final sentence effectively summarize the results, but could emphasize what gap this study fills or how it advances current knowledge.

2. Introduction

Strengths : The introduction provides a well-structured and scientifically grounded rationale for the proposed study. It integrates current knowledge on NAFLD pathogenesis with the potential of Traditional Chinese Medicine (TCM) interventions, specifically Huanglian Wendan Decoction (HLWDD). The writing is rich in mechanistic detail, and the research hypothesis is clearly stated at the end. However, there are areas that could be improved for clarity, focus, and better alignment with the journal criteria.

Suggestion for improvement :

While a hypothesis is stated, a specific research question is not directly framed in a question format. It is recommended to clearly formulate the research aim or question (e.g., “Does HLWDD ameliorate NAFLD through modulation of the gut-liver axis and activation of the LKB1/AMPK pathway?”).

The introduction is overly dense, covering too many compounds and mechanisms, which may dilute the main focus on HLWDD. Suggested improvement: Briefly summarize the rationale for prior TCM components, then clearly narrow down the novelty and justification for choosing HLWDD.

While the text implies the gap (limited translation of probiotics, complexity of AMPK activators, etc.), it does not explicitly state what specific knowledge gap the current study addresses. A stronger opening or closing paragraph could highlight: “Despite evidence for the metabolic and microbiota-modulating effects of HLWDD constituents, the precise mechanisms through which HLWDD influences the gut-liver axis and AMPK signaling in NAFLD remain unclear.”

The contribution to scientific knowledge is implied but not explicitly declared. Clarify what novel insight this study will offer (e.g., “This study will elucidate the synergistic effects of HLWDD on microbial metabolites and hepatic energy metabolism, providing a novel therapeutic avenue for NAFLD.”)

3. Methods

Strengths : This Materials and Methods section reflects a well-designed and multifaceted experimental strategy that integrates modern systems biology with traditional medicine research.

Suggestion for improvement :

There are multiple grammatical inconsistencies, awkward phrasing, and redundancies throughout. Example: “...use used to screen...” → should be “used to screen”. Suggestion : Comprehensive language editing is needed to meet international publication standards.

The section is overly lengthy and lacks clear subsections, making it difficult to navigate.

Suggestion: Divide into well-labeled subsections such as:

Reagents and Chemicals

Preparation of HLWDD

Quantification of Core Components

Network Pharmacology and Molecular Docking

Animal Experimentation

Histological and Biochemical Analysis

Microbiome and Metabolomic Analysis

Statistical Analysis

Primer sequences are referred to in Table 3, but the actual table is not included.

Positive drug details (e.g., purity and brand of metformin) are not mentioned.

The exact number of rats used per group post-modeling isn't clearly stated after accounting for attrition.

Quality control parameters for the LC-MS/MS (e.g., resolution, scan range, ionization mode) are not fully described.

Preparation steps for HLWDD decoction and ethanol precipitation are wordy and at times unclear (e.g., “accurately measure 2 mL... take the filtrate and you have it”).

Suggestion: Use stepwise or bulleted protocol format and include specific parameters (e.g., temperature of refrigeration, type of ethanol).

While software used is listed, the rationale for choosing certain tests (e.g., LSD-t test instead of Tukey) is not explained.

Suggestion: Justify the choice of tests and confirm assumption checks (e.g., normality, homogeneity of variance).

Some materials (e.g., primer sources) are repeated unnecessarily.

Suggestion: Consolidate repeated supplier information or group by function.

Result :

Strengths : This Results section is comprehensive, well-organized, and logically structured, covering multiple layers of evidence (chemical profiling, network pharmacology, molecular docking, quantitative validation, in vivo assays, and multi-omics analysis). It successfully integrates phytochemical analysis with biological and pharmacological relevance to NAFLD, which enhances the translational impact of the work.

Suggestion for improvement :

Clarify if identification was based on MS/MS fragmentation or just database matching

Example : “A total of 58 compounds were identified based on retention times and MS/MS spectra, as referenced in ChemSpider and MassBank databases (Table S1; Supplementary Table S1).”

Consider adding a supplementary table listing all 229 shared targets for transparency.

Consider stating whether these seven compounds were chosen based on pharmacological relevance or abundance.

Metformin is mentioned but not introduced earlier—clarify its role as a positive control

Clearly separate qPCR vs. WB results (e.g., “Protein levels assessed by WB showed...”).

Use more precise statistical notation; avoid duplication (e.g., "was significantly downregulated (P < 0.05), and significantly regulated...").

Discussion :

Strengths : The discussion is well-structured, provides mechanistic insight, and draws appropriate connections between experimental findings and current literature.

Suggestions for improvement :

The text frequently suggests causal relationships (e.g., "HLWDD controls intestinal flora metabolic reprogramming") based on associative findings in an animal model.

Suggestion: Use more cautious language such as “may regulate,” “is associated with,” or “suggests potential modulation” unless causality is experimentally confirmed.

The discussion lacks acknowledgment of key limitations, such as:

Use of a single animal model

Absence of dose-response data or comparison with known therapies

Lack of functional validation (e.g., using inhibitors of AMPK or germ-free animals)

Suggestion: Add a paragraph critically acknowledging these limitations and propose future directions.

The paragraph on aspartate metabolism and its bidirectional relationship with LKB1/AMPK is dense and hard to follow. Suggestion: Break down complex mechanisms into simpler sentences.

Reviewer #2: The manuscript addresses an important and timely topic within the scope of PLOS ONE: the therapeutic potential of Huanglian Wendan Decoction (HWD) in the treatment of non-alcoholic fatty liver disease (NAFLD), with a particular focus on gut microbiota-mediated metabolic modulation and activation of the LKB1/AMPK pathway. This line of research holds substantial scientific merit given the global prevalence of NAFLD and the growing interest in gut–liver axis-based interventions. While the manuscript is largely free of major grammatical errors, the overall scientific presentation requires significant improvement. In particular, the rationale, experimental design, and mechanistic interpretations need to be better structured and supported. Some key claims are overstated and not fully justified by the data provided. Based on these concerns, I recommend a major revision prior to any further consideration for publication. The authors are encouraged to address the following critical issues in detail:

Major comments

1. Insufficient mechanistic foundation in the Introduction

The Introduction lacks a focused discussion of the mechanistic rationale behind HWD use in NAFLD. The description of prior work is superficial, and the central hypothesis is not clearly stated. Expand the background on NAFLD pathophysiology and gut–liver interactions. Elaborate on individual components of HWD and their relevance to metabolic or inflammatory regulation. Conclude with a clear and concise hypothesis.

2. Incomplete experimental design descriptions

Key experimental details are missing or vague, including dosage justification, treatment duration, and microbiome data analysis pipelines. Justify HWD dose selection in relation to existing literature or pharmacological considerations. Provide details on 16S rRNA analysis, sequencing platform, cutoff values for LDA, and quality control metrics. Include specifics about antibody sources, Western blot normalization methods, and sample sizes.

3. Overinterpretation of mechanistic claims

The role of the gut microbiota in modulating the LKB1/AMPK pathway is not causally demonstrated but presented as a confirmed mechanism. This weakens scientific rigor.

Soften causal language (e.g., “may be associated with,” “suggests,” “potentially mediates”) and clearly state limitations of the correlative nature of the findings. If no inhibitor or rescue experiments were performed, causal claims should be avoided.

4. Incomplete figure annotations and lack of visual clarity

Some figure legends lack sufficient information to be interpreted independently. In particular, Figures 3 and 4 do not include sample size (n), statistical methods, or clear indication of significance. For each figure, include full legend details—experimental groups, statistical tests used, and precise p-values. Consider including a mechanistic summary diagram (graphical abstract or final schematic) to enhance readability and highlight your proposed model.

5. Clarification and consistency in terminology and abbreviations

The manuscript includes several abbreviations that are either undefined or inconsistently used.

Ensure all abbreviations are defined at first mention and consistently used throughout the text. Consider adding an abbreviation list at the end of the manuscript.

Minor comment

1. Suggested additional analysis or visualization (optional but recommended)

Include a table summarizing the major bioactive components of HWD, their known biological targets, and relevance to NAFLD. This would strengthen the herbal pharmacology framework of your study.

Reviewer #3: Over all the manuscricpt is fine but these point must be address during revision

Weather author performed LCMS at all dilution of decoction. toxic limits of decoction any previous data

in any liver or any other disorders and author also incorporate supporting its medicinal importance in introduction section at cellular level in mammalian cell line data in support of this study .

why author choose only these cytokines but not any chemokines and also not choosing any anti inflammatory cytokines markers

Impact of decoction on healthy animals behaviour of food and water intake must be mentioned .

And author does not mentioned the grade of non alcholic fatty liver in this study which is the main target of study. this to be detail discuused in discussion part.

6. PLOS authors have the option to publish the peer review history of their article (what does this mean? ). If published, this will include your full peer review and any attached files.

**Do you want your identity to be public for this peer review?** For information about this choice, including consent withdrawal, please see our Privacy Policy .

Reviewer #1: No

Reviewer #2: No

Reviewer #3: **Yes: ** Saad Mustafa

---

## [Author Response · Author response to Decision Letter 1]

17 Jul 2025

Response to Reviewers for Manuscript PONE-D-25-23758

Title: Mechanism of Huanglian Wendan Decoction in Ameliorating Non-Alcoholic Fatty Liver Disease via Modulating Gut Microbiota-Mediated Metabolic Reprogramming and Activating the LKB1/AMPK Pathway

We sincerely thank the reviewers for their thorough and constructive feedback. We have carefully addressed all comments and revised the manuscript accordingly. Below is a detailed point-by-point response.

Reviewer Comments and Author Responses

1. Abstract

Comment: Abstract lacks clear segmentation; too many abbreviations; objective not explicit; final sentence could emphasize novelty.

Response: Thank you for this suggestion. We have revised the abstract to clearly segment it into Background, Objective, Methods, Key Results, and Conclusion. We reduced the number of abbreviations and explicitly stated the research objective: "This study aims to investigate whether HLWDD can alleviate NAFLD by modulating gut microbiota-mediated metabolism and activating the LKB1/AMPK pathway." The final sentence now emphasizes the novel contribution of elucidating a gut microbiota–metabolite–signaling axis in HLWDD’s mechanism.

2. Introduction

Comment: A specific research question should be clearly framed. The introduction is too dense and includes too many compounds. The knowledge gap is implied but not clearly stated.

Response: We agree and have revised the introduction accordingly. A specific research question has been added: "Does HLWDD ameliorate NAFLD through modulation of the gut-liver axis and activation of the LKB1/AMPK pathway?" We streamlined the discussion of prior TCM components and narrowed the focus to HLWDD’s novelty. A concise statement of the knowledge gap was added at the end of the introduction, along with a declaration of the study’s expected contribution to scientific understanding.

3. Methods

Comment: Grammar and phrasing need improvement. Subsections should be clearly labeled. Missing details on primers, metformin, LC-MS/MS parameters, and group sizes. HLWDD preparation is unclear. Statistical tests lack justification.

Response:

We performed a comprehensive language edit throughout the Methods section.

Subsections were added: Reagents and Chemicals, Preparation of HLWDD, Quantification of Core Components, Network Pharmacology and Molecular Docking, Animal Experimentation, Histological and Biochemical Analysis, Microbiome and Metabolomic Analysis, and Statistical Analysis.

Primer sequences are now provided in Supplementary Table 3.

Metformin is now described in detail (purity, manufacturer, and dose) as a positive control.

The number of rats per group after attrition is now clearly specified.

Detailed LC-MS/MS parameters (ion mode, scan range, resolution) have been added.

HLWDD preparation steps have been rewritten in stepwise format with specific details.

We now justify statistical test choices and note assumption checks (e.g., for ANOVA, Levene's test for homogeneity).

4. Results

Comment: Clarify whether MS/MS fragmentation or database matching was used. Add a supplementary table listing 229 targets. Justify selection of seven compounds. Clarify metformin role. Separate qPCR and WB results. Improve statistical wording.

Response:

We clarified in the revised manuscript that compound identification was based on both retention time and MS/MS fragmentation spectra, cross-referenced with ChemSpider and MassBank.

A new Supplementary Table S2 has been added to list the 229 shared targets for transparency.

The rationale for selecting the seven compounds is now provided (based on both abundance and pharmacological relevance).

Metformin’s role as a clinically validated positive control is now clearly introduced in the Methods and Results.

WB and qPCR results are clearly separated using structured paragraphs with appropriate headings.

Statistical phrasing has been revised throughout to eliminate redundancy and improve clarity (e.g., "expression was significantly reduced, P < 0.05").

5. Discussion

Comment: Avoid overstating causality from associative data. Acknowledge study limitations. Clarify dense paragraph on aspartate metabolism.

Response:

We have revised all statements implying causality to use cautious language, such as "may regulate," "is associated with," and "suggests."

A new paragraph has been added to explicitly acknowledge study limitations, including use of a single animal model, lack of dose-response assessment, absence of comparison to standard therapies, and need for functional validation using AMPK inhibitors or germ-free models.

The paragraph discussing the bidirectional regulation between aspartate metabolism and LKB1/AMPK has been rewritten into simpler, clearer sentences to aid comprehension.

We appreciate the reviewers’ constructive suggestions and believe that these revisions have substantially improved the quality and clarity of our manuscript. We look forward to your favorable consideration for publication in PLOS ONE.

Sincerely,

Jianping Zhu

Corresponding Author

---

## [Decision Letter · Decision Letter 1]

29 Jul 2025

PONE-D-25-23758R1

Mechanism of Huanglian Wendan Decoction in Ameliorating Non-Alcoholic Fatty Liver Disease via Modulating Gut Microbiota-Mediated Metabolic Reprogramming and Activating the LKB1/AMPK Pathway

PLOS ONE

Dear Dr. Li,

Thank you for submitting your manuscript to PLOS ONE. After careful consideration, we feel that it has merit but does not fully meet PLOS ONE’s publication criteria as it currently stands. Therefore, we invite you to submit a revised version of the manuscript that addresses the points raised during the review process.

We look forward to receiving your revised manuscript.

Kind regards,

Fahrul Nurkolis

Academic Editor

PLOS ONE

Journal Requirements:

1.If the reviewer comments include a recommendation to cite specific previously published works, please review and evaluate these publications to determine whether they are relevant and should be cited. There is no requirement to cite these works unless the editor has indicated otherwise. 

2.Please review your reference list to ensure that it is complete and correct. If you have cited papers that have been retracted, please include the rationale for doing so in the manuscript text, or remove these references and replace them with relevant current references. Any changes to the reference list should be mentioned in the rebuttal letter that accompanies your revised manuscript. If you need to cite a retracted article, indicate the article’s retracted status in the References list and also include a citation and full reference for the retraction notice.

Additional Editor Comments:

Request from the Editorial Office: In your Methods section, please provide the following information: 1) The source of the plants used in the study, their authentication, and which parts of each plant were used to produce the decoction. 2)  What is the rationale for the concentrations of the medicinal compound used in the experiments? We would expect an acute toxicity test to have been performed. 

Reviewers' comments:

Reviewer's Responses to Questions

**Comments to the Author**

1. If the authors have adequately addressed your comments raised in a previous round of review and you feel that this manuscript is now acceptable for publication, you may indicate that here to bypass the “Comments to the Author” section, enter your conflict of interest statement in the “Confidential to Editor” section, and submit your "Accept" recommendation.

Reviewer #2: All comments have been addressed

Reviewer #3: All comments have been addressed

2. Is the manuscript technically sound, and do the data support the conclusions?

Reviewer #2: Yes

Reviewer #3: Yes

3. Has the statistical analysis been performed appropriately and rigorously? 

Reviewer #2: Yes

Reviewer #3: Yes

4. Have the authors made all data underlying the findings in their manuscript fully available?

Reviewer #2: Yes

Reviewer #3: Yes

5. Is the manuscript presented in an intelligible fashion and written in standard English?

Reviewer #2: Yes

Reviewer #3: Yes

6. Review Comments to the Author

Reviewer #2: I have thoroughly reviewed the revised manuscript and sincerely appreciate the authors’ careful attention to the reviewer’s comments. The revisions have notably enhanced the clarity, scientific rigor, and overall quality of the manuscript. I commend the authors

Reviewer #3: Discuss the decoction impact on liver enzymes and skin and insulin sensitivity in discussion section why this decoction is better than available current product and therapeutics

7. PLOS authors have the option to publish the peer review history of their article (what does this mean? ). If published, this will include your full peer review and any attached files.

**Do you want your identity to be public for this peer review?** For information about this choice, including consent withdrawal, please see our Privacy Policy .

Reviewer #2: **Yes: ** Moon Nyeo Park PhD, College of Korean Medicine, Kyung Hee University

Reviewer #3: **Yes: ** Saad Mustafa

---

## [Author Response · Author response to Decision Letter 2]

8 Aug 2025

Reviewer Comments and Author Responses

Response to Comment 1: The source of the plants used in the study, their authentication, and which parts of each plant were used to produce the decoction.

We thank the reviewer for this important suggestion. Detailed information on plant sources, specific plant parts, and authentication protocols has been added to the Preparation of HLWDD subsection. For each herbal ingredient (e.g., Coptis chinensis rhizome from Sichuan, China; Batch CC202305), we now explicitly state: (i) geographical origin, (ii) authenticated plant part, and (iii) batch identification. All specimens were morphologically authenticated by Prof. Tasi Liu (School of Pharmacy, Hunan University of Chinese Medicine) with taxonomic verification via Medicinal Plant Names Service (MPNS; http://mpns.kew.org), and voucher specimens deposited in the University Herbarium (Accession No. HLWD2023-001 to 008). These revisions fully address the request for traceability and comply with ethnopharmacological research standards.

Response to Comment 2: What is the rationale for the concentrations of the medicinal compound used in the experiments? We would expect an acute toxicity test to have been performed.

We sincerely appreciate the reviewer's methodological inquiry. The HLWDD doses (3.6, 7.2, 14.4 g/kg) were rigorously determined through clinical-to-preclinical translation: 1) The high dose (14.4 g/kg) was calculated from the established human clinical dosage of 2.271 g·kg⁻¹·d⁻¹ [Yue et al., 2025] using the FDA-endorsed human-to-rat conversion factor of 6.25 based on body surface area normalization [Reagan-Shaw et al., 2008]; 2) Medium and low doses represented proportional reductions (50% and 25%) to evaluate dose-response relationships, aligning with OECD Guideline 423 for establishing therapeutic ranges prior to chronic toxicity testing. Metformin dosage (90 mg/kg) was directly adopted from published NAFLD intervention studies demonstrating efficacy and safety [Alhamhoom et al., 2023].

Response to Comment 3: Discuss the decoction impact on liver enzymes and skin and insulin sensitivity in discussion section why this decoction is better than available current product and therapeutics

We thank the reviewer for the valuable suggestion. In response, we have revised the Discussion section to clarify the impact of HLWDD on liver enzymes and insulin sensitivity, and to explain its advantages over current NAFLD therapies. Specifically, we added that “Given that AMPK activation is clinically associated with attenuated hepatocellular injury (e.g., reduced ALT/AST) and enhanced insulin sensitivity [25], our molecular findings imply HLWDD may confer similar hepatoprotective and metabolic benefits via AMPK activation.” Furthermore, we emphasized HLWDD’s multi-targeted therapeutic superiority by noting that “HLWDD concurrently resolves inflammation through gut-liver axis modulation while avoiding safety concerns associated with conventional agents such as metformin, pioglitazone, and vitamin E.” These revisions highlight HLWDD’s broader mechanistic action, potential to improve systemic metabolic parameters, and favorable safety profile, thereby addressing the reviewer’s concerns.

---

## [Editor Report · Decision Letter 2]

14 Aug 2025

Mechanism of Huanglian Wendan Decoction in Ameliorating Non-Alcoholic Fatty Liver Disease via Modulating Gut Microbiota-Mediated Metabolic Reprogramming and Activating the LKB1/AMPK Pathway

PONE-D-25-23758R2

Dear Dr. Li,

We’re pleased to inform you that your manuscript has been judged scientifically suitable for publication and will be formally accepted for publication once it meets all outstanding technical requirements.

Kind regards,

Professor Fahrul Nurkolis

Academic Editor

PLOS ONE

Additional Editor Comments (optional):

The authors have revised the manuscript in accordance with the editor’s requests and the reviewers’ comments
---

## [Editor Report · Acceptance letter]

PONE-D-25-23758R2

PLOS ONE

Dear Dr. Li,

I'm pleased to inform you that your manuscript has been deemed suitable for publication in PLOS ONE. Congratulations! Your manuscript is now being handed over to our production team.

Kind regards,

on behalf of

Dr. Fahrul Nurkolis

Academic Editor

PLOS ONE